# AKT isoforms have distinct hippocampal expression and roles in synaptic plasticity

Josien Levenga[1,2], Helen Wong[1], Ryan A Milstead[3], Bailey N Keller[1], Lauren E LaPlante[1], Charles A Hoeffer[1,2,3]*

[1]Institute for Behavioral Genetics, University of Colorado-Boulder, Boulder, United States; [2]Linda Crnic Institute, Aurora, United States; [3]Department of Integrative Physiology, University of Colorado-Boulder, Boulder, United States

**Abstract** AKT is a kinase regulating numerous cellular processes in the brain, and mutations in *AKT* are known to affect brain function. AKT is indirectly implicated in synaptic plasticity, but its direct role has not been studied. Moreover, three highly related AKT isoforms are expressed in the brain, but their individual roles are poorly understood. We find in *Mus musculus*, each AKT isoform has a unique expression pattern in the hippocampus, with AKT1 and AKT3 primarily in neurons but displaying local differences, while AKT2 is in astrocytes. We also find isoform-specific roles for AKT in multiple paradigms of hippocampal synaptic plasticity in area CA1. AKT1, but not AKT2 or AKT3, is required for L-LTP through regulating activity-induced protein synthesis. Interestingly, AKT activity inhibits mGluR-LTD, with overlapping functions for AKT1 and AKT3. In summary, our studies identify distinct expression patterns and roles in synaptic plasticity for AKT isoforms in the hippocampus.

DOI: https://doi.org/10.7554/eLife.30640.001

*For correspondence:
charles.hoeffer@colorado.edu

Competing interests: The authors declare that no competing interests exist.

## Introduction

Dysregulation of synaptic plasticity is implicated in cognitive and memory impairments associated with many neurological diseases and psychiatric disorders, such as Alzheimer's disease, schizophrenia, and intellectual disability. The AKT (Protein kinase B, PKB) signaling pathway is thought to play a pivotal role in synaptic plasticity (*Hers et al., 2011*). AKT is a central serine/threonine kinase expressed in almost all cell types throughout the body and regulates cell growth, proliferation and metabolism. In post-mitotic neurons, the AKT pathway has significant functional impact on stress responses, neurotransmission and synaptic plasticity (*O' Neill, 2013*).

Synaptic plasticity describes the specific modification of neuronal connections in response to neural activity. A persistent increase in synaptic strength after stimulation is known as long-term potentiation (LTP), while a decrease in synaptic strength is known as long-term depression (LTD). Late-phase LTP (L-LTP) is a long-lasting form of LTP that requires changes in gene expression and the synthesis of new proteins (*Sweatt, 2016*). A form of LTD mediated by group one metabotropic glutamate receptors (mGluR-LTD) also requires protein synthesis (*Huber et al., 2000*). Interestingly, AKT plays an important role in the mammalian target of rapamycin (mTOR) pathway controlling protein synthesis, and previous studies found that induction of L-LTP or mGluR-LTD in the hippocampus leads to AKT activation measured by increased phosphorylation of serine 473 (*Hou and Klann, 2004*; *Horwood et al., 2006*).

AKT is known to include a family of three closely related isoforms, named AKT1, AKT2 and AKT3. Each isoform is encoded by a different gene, but the proteins share a high degree of structural homology (*Kumar and Madison, 2005*). Despite this homology, accumulating evidence suggests that each isoform is involved in distinct neurological disorders. Mutations in *AKT1* have been associated with schizophrenia, *AKT2* with gliomas and *AKT3* with brain growth (*Emamian et al., 2004*;

*Zhang et al., 2010*; *Lee et al., 2012*). Supporting these observations in humans, single-isoform *Akt* knockout (KO) mice also show distinct phenotypes. *Akt1* KO mice have growth retardation and increased neonatal death (*Easton et al., 2005*; *Yang et al., 2005*), *Akt2* KO mice suffer from a type two diabetes-like syndrome and *Akt3* KO mice show decreased brain size (*Easton et al., 2005*; *Cho et al., 2001a*; *Tschopp et al., 2005*). These data suggest that each isoform subserves different cellular functions to give rise to distinct phenotypes. Whether each AKT isoform plays different roles in synaptic plasticity has not been examined.

Activation of AKT has been correlated with LTP and LTD induction, implicating a role for AKT (*Hou and Klann, 2004*; *Horwood et al., 2006*; *Nakai et al., 2014*). However, whether AKT activity is necessary in synaptic plasticity has not been directly tested. Moreover, all three AKT isoforms are present in the brain (*Easton et al., 2005*). Given the numerous forms of synaptic plasticity, isoform-specific functions of AKT may provide an important mechanism for control and precision of the cellular processes supporting synaptic plasticity. Here, we show for the first time that each AKT isoform has a distinct expression pattern in the hippocampus. We then examined the role of each isoform in several hippocampal synaptic plasticity paradigms known to involve different molecular and cellular processes: early-phase LTP (E-LTP), L-LTP, low-frequency stimulation (LFS)-LTD, and mGluR-LTD. Our studies provide evidence that AKT isoforms play differential roles in synaptic plasticity due to cell-type-specific expression of *Akt* genes in the hippocampus and isoform-specific functions in protein synthesis.

## Results

### AKT isoforms show differential expression in the hippocampus

AKT is a well-studied kinase that may play a central role in brain disorders (*Hers et al., 2011*). However, most studies examining AKT activity made no distinction between the activities of each isoform. We hypothesized that AKT isoforms play distinct roles in synaptic plasticity. To test this, we first examined the expression pattern of AKT1, AKT2 and AKT3 in the mouse hippocampus. We found that AKT1 and AKT3 were distributed throughout somatic layers of the hippocampus, with local differences in expression (*Figure 1*). AKT1 showed more intense immunoreactivity in the cell body layer of area CA1, whereas AKT3 showed more intense staining in cell bodies within area CA3 and the hilus of the dentate gyrus but reduced expression in the CA2/CA1 region. Interestingly, AKT2 showed a very different staining pattern. AKT2 was mostly expressed in cells of the molecular layer of the dentate gyrus and stratum radiatum of CA1 (*Figure 1*). Isoform-specific KO tissues confirmed specificity of the staining (*Figure 1*). Because the brain consists of neurons and glia cells, we next determined the cell types in which each isoform is expressed by co-staining brain slices for the neuronal marker neuron-specific nuclear protein (NeuN), the astrocytic marker glial fibrillary acidic protein (GFAP) and each AKT isoform. This triple stain approach revealed that within CA1, AKT1 was mainly expressed in pyramidal layer neurons, with no detectable staining in astrocytes (*Figure 2a,b*). Interestingly, certain neurons seemed to express more AKT1 compared with neighboring neurons (*Figure 2a,b*). In the CA1, AKT3 was also mainly expressed in neurons, including in the processes extending into stratum radiatum (*Figure 2a,b*). AKT2 showed no detectable co-localization with NeuN but co-localized with GFAP (*Figure 2a,b*). To confirm that AKT2 is not expressed in neurons, we employed a Cre-mediated strategy of selective *Akt2* gene disruption by crossing mice with floxed alleles of *Akt2* to three different mouse lines. Two of the lines have neuron-specific Cre recombinase expression (*Camk2α*-Cre and rat specific enolase, Nse or *Eno2*-Cre), while the third line expresses Cre recombinase in early neural progenitor cells that become astrocytes and neurons (*Nestin*-Cre [*Tronche et al., 1999*]). Using this strategy, we confirmed that AKT2 levels in the hippocampus were not affected by either neuron-specific Cre line, showing that *Akt2* is not likely to be expressed in hippocampal neurons (*Figure 2—figure supplement 1a*). In contrast, the Nestin-Cre line completely abolished hippocampal AKT2 expression (*Figure 2—figure supplement 1b,c*), providing further support that AKT2 protein may be solely expressed in astrocytes and not in neurons of the hippocampus. Combined, our results demonstrate differential AKT isoform expression in the hippocampus. This isoform-specific expression may lead to unique functions of each isoform in synaptic plasticity processes.

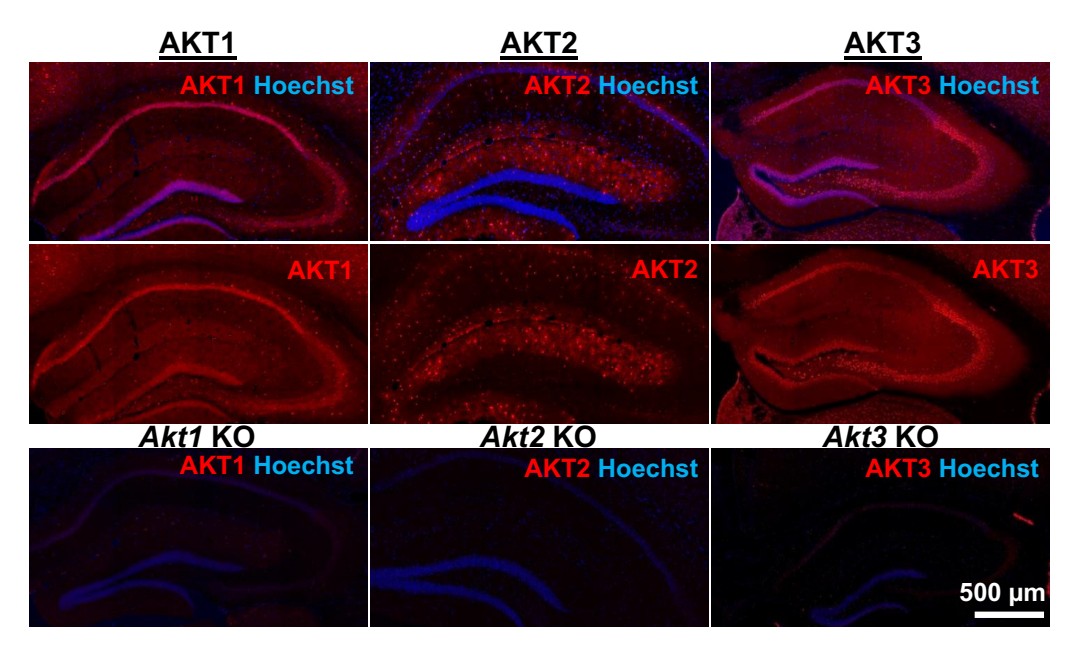

**Figure 1.** AKT isoform-specific expression in the hippocampus. Immunohistology using isoform-specific antibodies revealed distinct expression patterns for each AKT isoform in the hippocampus. AKT1 was mainly expressed in the cell body layers, with the greatest levels in stratum pyramidale of CA1. AKT2 was mostly expressed in specific cells in the molecular layer of the dentate gyrus, CA3 and CA1. AKT3 was also mainly expressed in the cell body layers of the hippocampus and showed strong expression in the hilus and CA3. Bottom panels show single *Akt* knockout (KO) tissue to validate the specificity of the antibodies.

DOI: https://doi.org/10.7554/eLife.30640.002

## AKT inhibition leads to reduced substrate activity and impaired L-LTP

AKT has been implicated in L-LTP (*Huang et al., 2013*), but directly targeting AKT in L-LTP had not been studied. We chose to test the effects of two AKT inhibitors, AZD5363 (AZD) and MK2206-HCl (MK), on synaptic plasticity. AZD and MK have different mechanisms of action; AZD is an ATP-competitive AKT inhibitor (*Zhang et al., 2016*) and MK is an allosteric AKT inhibitor (*Liu et al., 2011*). To examine AZD and MK action in the brain, we first assessed their effects on the phosphorylation of AKT and a well-characterized downstream target of AKT, glycogen synthase kinase 3β (GSK-3β) in hippocampal slices. Incubation with 10 μM MK strongly inhibited AKT activity as measured by the reduction of AKT phosphorylated on serine 473 (pAKT S473); however, phosphorylation levels of GSK-3β on serine 9 (pGSK-3β S9) were not reduced (*Figure 3a–c*). At higher doses of MK (30 μM and 100 μM), pAKT continued to be reduced at similar levels (*Figure 3a,b*), while pGSK-3β levels were ultimately reduced in a dose-dependent manner (*Figure 3a,c*). MK also reduced phosphorylation of another AKT target, tuberous sclerosis complex 2 on serine 939 (pTSC2 S939) (*Inoki et al., 2002*) but did not have a dose-dependent effect (*Figure 3a,d*). Additionally, we investigated the effect of MK on individual AKT isoforms and found that MK inhibits all isoforms, (*Figure 3—figure supplement 1*).

Incubation of AZD, the other AKT inhibitor, resulted in increased pAKT levels collectively (*Figure 3a,b*) as well as individually for all three AKT isoforms (*Figure 3—figure supplement 1*). Although seemingly paradoxical, this effect is consistent with previous reports that ATP-competitive inhibitors of AKT result in hyperphosphorylation of the kinase itself. The hyperphosphorylation does not indicate increased AKT activation but rather reflects intrinsic responses to competitive inhibition of AKT (*Okuzumi et al., 2009*). Previous studies have shown that binding of ATP-competitive AKT inhibitors results in translocation of AKT to the membrane where AKT is either more susceptible to phosphorylation (*Okuzumi et al., 2009*) or more resistant to dephosphorylation (*Chan et al., 2011*), thereby leading to AKT hyperphosphorylation. To demonstrate that there is no feedback on upstream signals to compensate for AKT inhibition in brain slices, we examined phosphorylation of

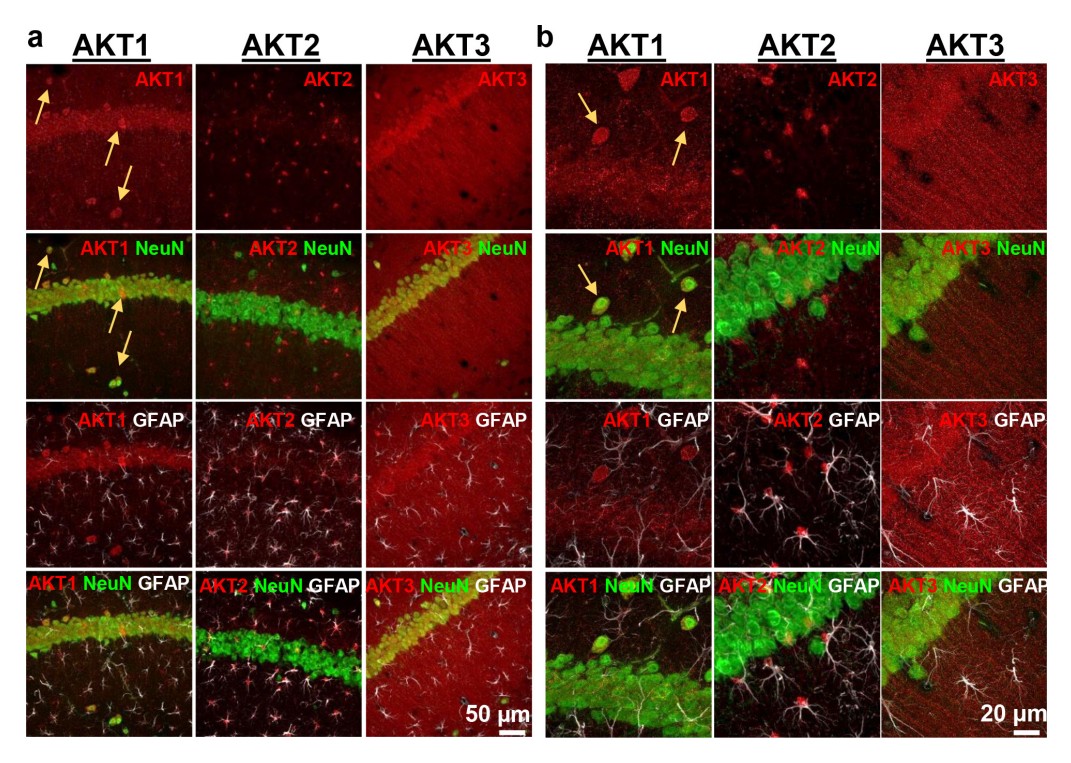

**Figure 2.** Cell-type-specific expression of AKT isoforms in hippocampal area CA1. (**a**). AKT1 was mainly expressed in neuronal cell bodies, indicated by co-localization with NeuN. Certain neurons in the pyramidal layer, stratum oriens and molecular layer showed greater expression levels of AKT1 (yellow arrows). AKT2 was specifically expressed in astrocytes, shown by co-localization with the astrocyte marker GFAP. AKT3 co-localized with NeuN like AKT1 but was also expressed in the stratum radiatum, most likely within dendrites. (**b**) Higher magnification showed no specific co-localization of AKT1 with GFAP and that certain neurons in the stratum oriens expressed high levels of AKT1 (yellow arrows). AKT2 was mainly expressed in the cell bodies of astrocytes. AKT3 showed high expression levels in neuronal cell bodies and dendrites and some expression in astrocytes.

DOI: https://doi.org/10.7554/eLife.30640.003

The following figure supplement is available for figure 2:

**Figure supplement 1.** Cell-type-specific expression of AKT2 in the hippocampus.

DOI: https://doi.org/10.7554/eLife.30640.004

3-phophoinositide-dependent protein kinase-1 (PDK1), a direct upstream activator of AKT, and found no effect of AZD on PDK1 activation (**Figure 3—figure supplement 1a**). To confirm that AZD inhibited AKT activity despite the hyperphosphorylation of AKT, we examined AKT substrates and found significantly decreased phosphorylation levels of both GSK-3β, in a dose-dependent manner, and TSC2 (**Figure 3a,c,d**). Finally, the AKT1/2 inhibitor A6730 (10 mM) was recently found to reduce levels of the AMPA receptor subunit GluA2 (**Pen et al., 2016**), but we found no reduction even with the highest concentration of AZD or MK (**Figure 3a,d**).

Based on the dose-response reduction of pGSK-3β S9 levels to AZD and MK, we chose to test 30 μM for both AKT inhibitors on synaptic plasticity in the CA3-CA1 circuit of the hippocampus. After determining that incubation of hippocampal slices from wild-type (WT) mice with either inhibitor did not affect the stability of baseline field excitatory postsynaptic potentials (fEPSP) recorded in CA1 up to 90 min after incubation (**Figure 4—figure supplement 1**), we assessed the effect on expression of L-LTP induced by high-frequency stimulation (HFS). We found that both AKT inhibitors resulted in significantly impaired L-LTP compared with vehicle treatment (**Figure 4a,b**), indicating that AKT activity is required to sustain long-lasting LTP in area CA1.

## Loss of AKT1 leads to impaired L-LTP

To determine if a specific AKT isoform was responsible for maintaining L-LTP in the CA3-CA1 circuit, we examined hippocampal slices from *Akt1*, *Akt2* or *Akt3* KO mice (**Figure 4c**). Because AKT has

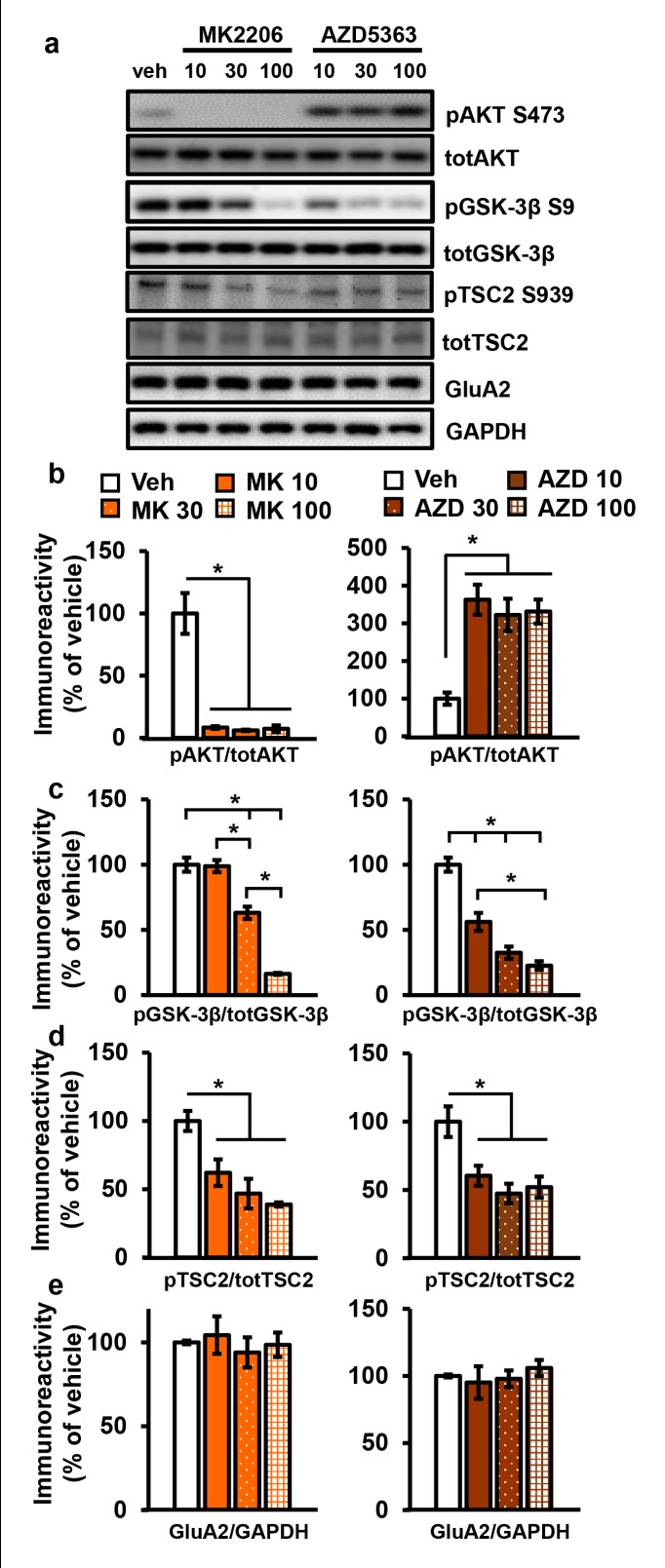

**Figure 3.** AKT inhibitors MK2206 and AZD5363 effectively inhibit AKT activity in hippocampal slices. (**a**) MK2206 and AZD5363 blocked AKT activity effectively, shown by reduced phosphorylation levels of the AKT substrate GSK-3β on serine 9 (pGSK-3β S9), but had different effects on phosphorylation levels of AKT itself on serine 473 (pAKT S473). (**b**) MK2206 (MK) significantly reduced pAKT S473 levels normalized to total AKT (totAKT) levels,

*Figure 3 continued on next page*

*Figure 3 continued*

while AZD5363 (AZD) significantly increased pAKT S473 levels (MK $F_{(3,8)}$ = 30.93, p<0.0001; AZD $F_{(3,8)}$ = 12.30, p=0.0023). (c) MK had a dose-dependent effect on pGSK-3β S9 levels normalized to total GSK-3β (totGSK-3β) levels, with no effect at 10 µM but significantly reduced levels at 30 and 100 µM compared with vehicle treatment (MK $F_{(3,8)}$=74.56, p<0.001; post hoc comparisons: veh vs. MK10 p=0.998; veh vs. MK30 p=0.002; veh vs. MK100 p<0.001, MK10 vs. MK30 p=0.002, MK30 vs. MK100 p<0.001). AZD also had a dose-dependent effect on pGSK3β-S9 levels, with significant reduction at 10 µM in addition to the 30 and 100 µM doses compared with vehicle (AZD $F_{(3,8)}$=39.69, p<0.001; post hoc comparisons: veh vs. AZD10 p=0.002; veh vs. AZD30 p<0.001; veh vs. AZD100 p<0.001; AZD10 vs. AZD100 p=0.01). (d) MK and AZD treatment significantly reduced phosphorylation of TSC2 at serine 939 (pTSC2 S939) (MK $F_{(3,9)}$=10.27, p=0.0041; AZD $F_{(3,9)}$=7.893, p=0.0089). (e) Levels of GluA2 normalized to GAPDH was not significantly altered after MK or AZD treatment (MK $F_{(3, 8)}$=0.279, p=0.838; AZD $F_{(3,9)}$=0.378, p=0.771). Veh = vehicle, MK10 = 10 µM, MK30 = 30 µM, MK100 = 100 µM; AZD10 = 10 µM, AZD30 = 30 µM, AZD100 = 100 µM, 3–4 mice/group, *p<0.05.

DOI: https://doi.org/10.7554/eLife.30640.005

The following figure supplement is available for figure 3:

**Figure supplement 1.** Effect of MK2206 or AZD5363 on phosphorylation of AKT isoforms and PDK1 in the hippocampus.

DOI: https://doi.org/10.7554/eLife.30640.006

been linked to AMPA and NMDA receptor function (*Pen et al., 2016*), we first examined the effect of deleting single *Akt* isoforms on basal synaptic transmission, presynaptic plasticity and short-term LTP in area CA1. We found normal synaptic input/output curves (*Figure 4—figure supplement 2a–c*), paired-pulse facilitation (*Figure 4—figure supplement 2d–f*) and E-LTP (*Figure 4—figure supplement 3*) in all single *Akt* isoform mutants, suggesting no one isoform is required for normal basal synaptic transmission, presynaptic plasticity and short-term LTP. In contrast, we found that *Akt1* deletion resulted in impaired L-LTP (*Figure 4d*), while slices from *Akt2* or *Akt3* KO mice showed normal L-LTP (*Figure 4e,f*). To test if these mutant mice exhibit AKT isoform compensation, we examined the phosphorylation status of AKT3 in *Akt1* KO mice and, conversely, AKT1 in *Akt3* KO mice. We found no significant effect on pAKT3 in *Akt1* KO mice, while *Akt3* KO mice showed a small but significant increase in pAKT1 (*Figure 4—figure supplement 4*). This suggests that AKT3 is unable to compensate for AKT1 loss in L-LTP, resulting in impaired L-LTP in *Akt1* KO mice, while AKT1 may be able to compensate for AKT3 loss, allowing normal L-LTP in *Akt3* KO mice. Together, these results show that AKT1 is the major isoform involved in the expression of L-LTP and targeted by AZD and MK inhibition to impair L-LTP in CA1.

### *Akt1* deletion results in impaired protein synthesis after L-LTP induction

Hippocampal L-LTP requires synthesis of proteins that are involved in modifying synaptic connections (*Sweatt, 2016*). Because the AKT pathway plays a well-known role in translational control, the AKT1 isoform may function in L-LTP by regulating protein synthesis. Interestingly, even though *Akt3* removal had no effect on L-LTP, *Akt3* KO mice have significantly smaller brains, which may indicate impaired translation. To examine protein synthesis associated with L-LTP in *Akt1* and *Akt3* KO hippocampal slices, we used the SuNSET method of labeling newly synthesized proteins (*Hoeffer et al., 2013*). Slices from *Akt1* or *Akt3* KO and WT littermates received either four spaced trains of HFS to induce L-LTP or no stimulation (control). Western blotting showed that *Akt1* KO slices failed to increase protein synthesis following L-LTP induction (*Figure 5a,c*), consistent with the L-LTP impairment observed in these slices (*Figure 4d*). In contrast, *Akt3* deletion did not affect the protein synthesis response post-HFS (*Figure 5b,d*), which is consistent with the normal L-LTP found in *Akt3* KO slices (*Figure 4f*). These results suggest an isoform-specific role for AKT1 in activity-induced protein synthesis that promotes L-LTP expression in the hippocampus.

To investigate the molecular signaling pathways linking AKT activity to translational control in long-lasting synaptic plasticity, we examined the phosphorylation levels of a well-known AKT effector, S6. S6 is a component of the 40S ribosomal subunit that mediates translation, and phosphorylation of S6 at serine 235 and 236 (pS6) is essential for S6 cap-binding activity (*Hutchinson et al., 2011*). We found that pS6 levels significantly increased after tetanic stimulation in WT and *Akt3* KO hippocampal slices but not in *Akt1* KO slices (*Figure 5e–h*). This result agrees with the impaired

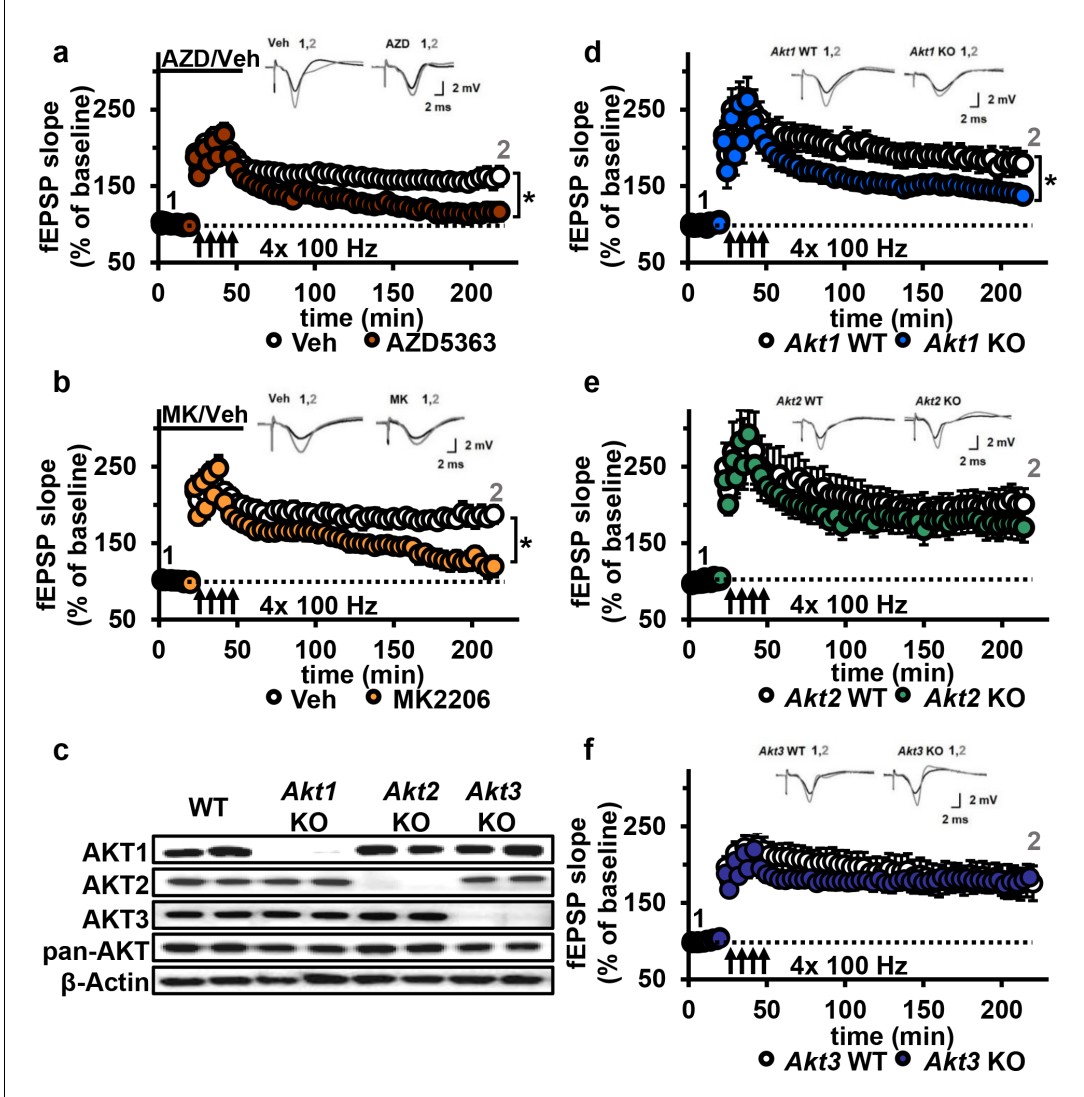

**Figure 4.** AKT inhibition and *Akt1* deletion results in impaired L-LTP. (a) Inhibiting AKT activity in hippocampal slices with AZD5363 (30 µM) prior to four trains of HFS (100 Hz) resulted in significantly impaired L-LTP in hippocampal area CA1 compared with vehicle treatment ($F_{(1,24)}$=6.896, p=0.015), $n$ = 13 slices/group, 5 mice/group. (b) AKT inhibitor MK2206 (30 µM) also significantly impaired L-LTP compared with vehicle treatment ($F_{(1,13)}$=18.639, p<0.001), $n$ = 7–8 slices/group, 4 mice/group. (c) Western blot validation of AKT1, AKT2 and AKT3 removal using cortical tissue from *Akt* isoform-specific KO mice. β-actin, loading control. (d) *Akt1* KO mice showed significantly impaired L-LTP in hippocampal area CA1 compared with WT mice ($F_{(1,28)}$=5.049, p=0.033), $n$ = 15 slices/group, 6 mice/group. (e) *Akt2* KO mice displayed similar levels of L-LTP to WT mice ($F_{(1,20)}$=1.121, p=0.302), $n$ = 10–12 slices/group, 4–5 mice/group. (f) *Akt3* KO mice showed similar levels of L-LTP to WT mice ($F_{(1,20)}$=0.33, p=0.857), $n$ = 8–14 slices/group, 5–7 mice/group. *p<0.05.

DOI: https://doi.org/10.7554/eLife.30640.007

The following figure supplements are available for figure 4:

**Figure supplement 1.** Effect of MK2206 or AZD5363 on basal fEPSPs in the hippocampus.

DOI: https://doi.org/10.7554/eLife.30640.008

**Figure supplement 2.** Deletion of single Akt isoforms has no effect on basal transmission or presynaptic plasticity.

DOI: https://doi.org/10.7554/eLife.30640.009

**Figure supplement 3.** Deletion of single Akt isoforms have no effect on E-LTP.

DOI: https://doi.org/10.7554/eLife.30640.010

**Figure supplement 4.** Single isoform phosphorylation in *Akt1* and *Akt3* KO mice.

DOI: https://doi.org/10.7554/eLife.30640.011

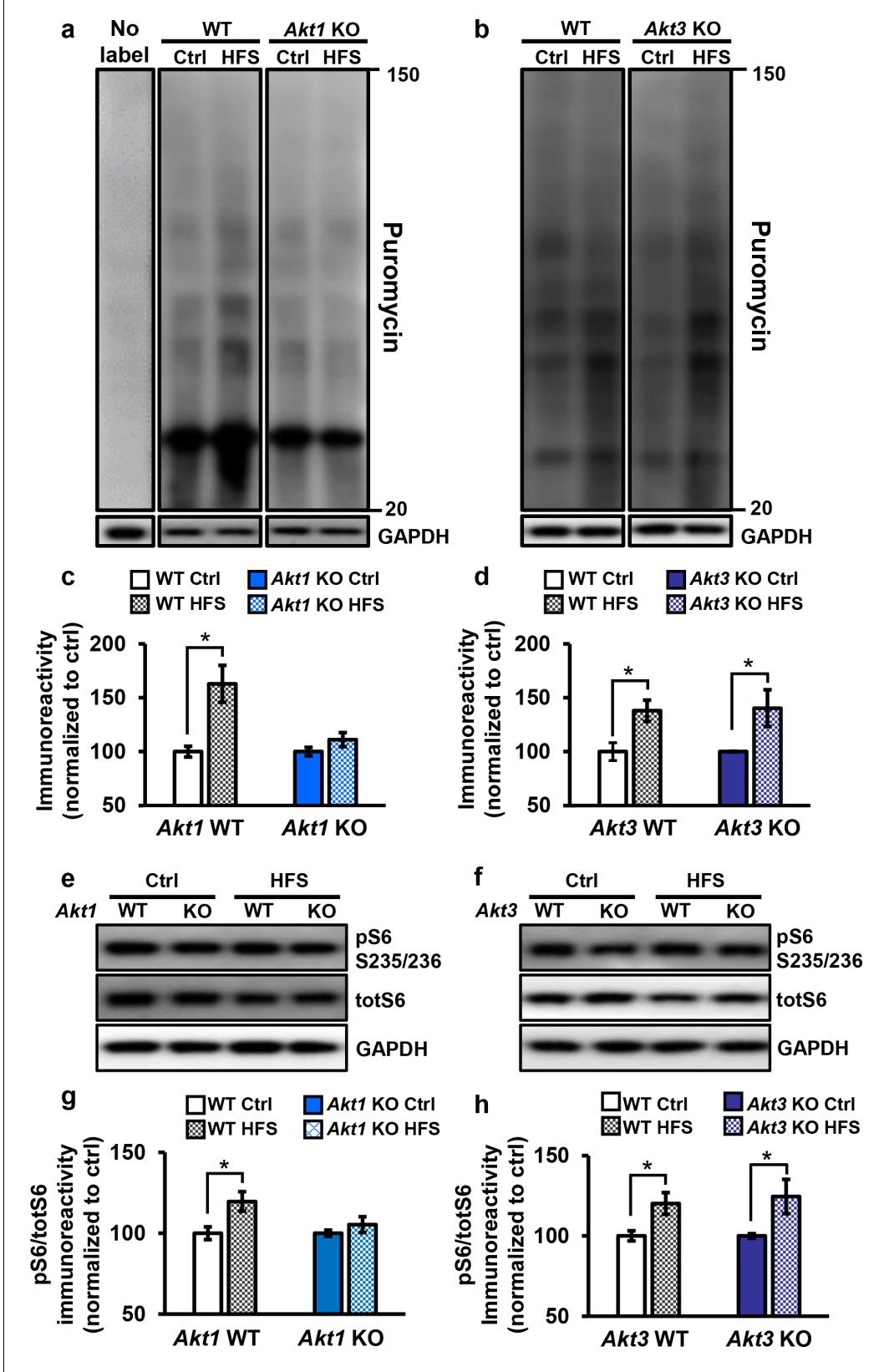

**Figure 5.** Only *Akt1* KO mice show an impaired protein synthesis response after tetanic stimulation. (**a,c**) Puromycin labeling of newly synthesized proteins showed that four trains of HFS in area CA1 to induce L-LTP results in increased protein synthesis levels compared with no stimulation (Ctrl) in

*Figure 5 continued on next page*

Figure 5 continued

WT hippocampal slices ($t_{(14)}$= −3.52, p=0.005), while stimulated *Akt1* KO slices fail to increase protein synthesis from unstimulated levels ($t_{(14)}$=-1.45, p=0.167). GAPDH, loading control. (**b,d**) *Akt3* KO hippocampal slices showed a normal increase in protein synthesis after four trains of HFS (*Akt3* WT Ctrl vs. HFS $t_{(12)}$=-2.75, p=0.018; *Akt3* KO Ctrl vs. HFS t = −2.20, p=0.047). GAPDH, loading control. (**e,g**) *Akt1* KO slices failed to show an increase in S6 phosphorylation (pS6 S235/236) normalized to total S6 (totS6) levels after tetanic stimulation (*Akt1* WT Ctrl vs. HFS $t_{(14)}$=-2.71, p=0.016; *Akt1* KO Ctrl vs. HFS $t_{(14)}$=-1.01, p=0.33). (**f,h**) *Akt3* KO slices showed a normal increase in pS6 S235/236 levels after tetanic stimulation (*Akt3* WT Ctrl vs. HFS $t_{(14)}$=-2.92, p=0.01, *Akt3* KO Ctrl vs. HFS $t_{(14)}$=-2.59, p=0.02). n= 7–11 slices/group, 4–6 mice/group, *p<0.05.
DOI: https://doi.org/10.7554/eLife.30640.012

protein synthesis after tetanic stimulation observed with AKT1 deficiency. Therefore, AKT1 is the critical isoform supporting hippocampal L-LTP through activity-regulated signaling to protein synthesis and may compensate for AKT3 to enable normal L-LTP and associated protein synthesis in *Akt3* KO conditions.

## AKT activity is not required for LTD induced by low-frequency stimulation

Another form of synaptic plasticity is LTD induced by low-frequency stimulation (LFS-LTD), which is known to depend on NMDA receptor-mediated calcium influx and is independent of protein synthesis (*Huber et al., 2000*). Previously, synaptic plasticity related to brain-derived neurotrophic factor signaling was reported to involve a feedforward AKT mechanism to NMDA receptors (*Nakai et al., 2014*). Furthermore, the AKT substrate GSK-3β was found to be important for LFS-LTD, and phosphorylation levels of GSK-3β was found to be reduced in *Akt3* KO brains (*Peineau et al., 2007*; *Bergeron et al., 2017*). We confirmed that *Akt3* deletion results in decreased hippocampal levels of pGSK-3β (*Figure 6a*). In contrast, *Akt1* KO mice show normal pGSK-3β levels (*Figure 6a*), revealing further differential AKT isoform signaling. Therefore, AKT activity, especially AKT3, may be involved in maintaining LFS-LTD. To address this question, we examined LFS-LTD in the CA3-CA1 circuit of single *Akt* isoform mutant slices and found no effect of any single isoform deletion (*Figure 6b–d*). To determine if these findings resulted from incomplete blockade of AKT activity or if multiple isoforms are involved, we examined LFS-LTD in WT hippocampal slices following pharmacological inhibition of AKT activity using AZD or MK. We found normal expression of LFS-LTD compared to vehicle-treated slices (*Figure 6e,f*). Therefore, these experiments combined indicate that AKT is not required for LFS-LTD in CA1.

## Inhibition of AKT activity leads to enhanced mGluR-LTD

Previous studies suggest AKT is involved in another form of LTD, which is mediated by mGluR (mGluR-LTD) and depends on protein synthesis (*Huber et al., 2001*). Upon mGluR stimulation with DHPG, pAKT S473 levels were shown to increase, suggesting increased AKT activity (*Hou and Klann, 2004*). Also, inhibiting phosphoinositide 3-kinase (PI3K) activity, a well-validated kinase upstream of AKT activation (*Frech et al., 1997*), results in impaired mGluR-LTD (*Hou and Klann, 2004*). However, direct interrogation of the role of AKT in mGluR-LTD expression had never been performed. Thus, we directly examined AKT function in mGluR-LTD using the pan-AKT inhibitors AZD and MK and *Akt* mutants.

We incubated WT hippocampal slices with AKT inhibitors prior to mGluR-LTD induction with DHPG (*Ito et al., 1992*). Unexpectedly, we found that inhibiting AKT activity with either AZD or MK enhanced mGluR-LTD (*Figure 7a,b*). Because a previous report using LY294002 inhibition of PI3K, a major upstream activator of AKT (*Wymann et al., 2003*), showed impaired mGluR-LTD (*Hou and Klann, 2004*), we had expected to find that direct AKT inhibition also would impair mGluR-LTD (*Hou and Klann, 2004*). We therefore repeated the experiment with LY294002. In agreement with our MK and AZD data, we found that inhibition of PI3K resulted in enhanced mGluR-LTD (*Figure 7c*). These results demonstrate that AKT is indeed involved in regulating mGluR-LTD, although in a different manner than we expected.

To investigate isoform-specific contributions to this phenotype, we next induced mGluR-LTD in *Akt1*, *Akt2* or *Akt3* KO hippocampal slices. Interestingly, we found that all single isoform mutant slices displayed normal mGluR-LTD (*Figure 7d–f*). Because pan-AKT and PI3K inhibition both resulted in enhanced mGluR-LTD (*Figure 7a–c*), two or more isoforms may redundantly compensate for each

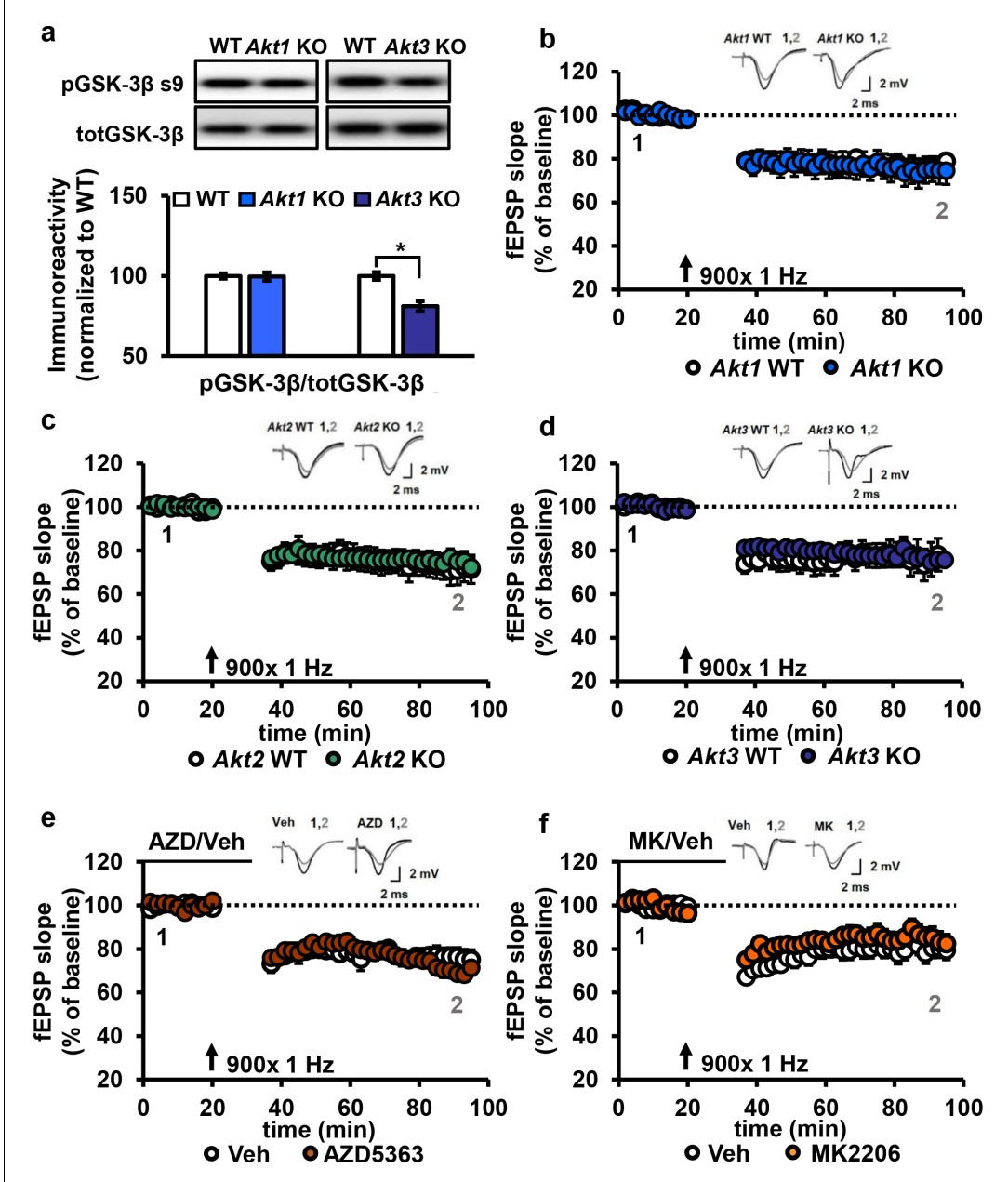

**Figure 6.** AKT is not involved in LFS-LTD. (a) pGSK-3β levels were significantly reduced in the hippocampus of *Akt3* KO mice but normal in *Akt1* KO mice (*Akt3* KO $t_{(20)}$=-4.725, p<0.001; *Akt1* KO $t_{(14)}$=0.104, p=0.918), 4–6 mice/group. (b–d) Hippocampal slices from *Akt1* KO, *Akt2* KO or *Akt3* KO mice showed normal LFS-LTD induced by 900 stimuli of 1 Hz compared to WT slices (p>0.05 for all genotypes), n = 12–20 slices/group, 5–7 mice/group. (e) Blocking AKT activity in hippocampal slices with AZD5363 (30 μM) prior to 900 stimuli of 1 Hz did not affect LFS-LTD compared to vehicle-treated slices (p>0.05), n = 12–13 slices/group, 6 mice/group. (f) AKT activity blocked by MK2206 (30 μM) also resulted in normal LFS-LTD in hippocampal slices compared to vehicle-treated slices (p>0.05), n = 10–12 slices/group, 5 mice/group.

DOI: https://doi.org/10.7554/eLife.30640.013

other to allow normal expression of mGluR-LTD in the single *Akt* isoform mutants. To test this idea, we generated multiply mutant *Akt* mice.

### *Akt1/Akt3* double mutants have enhanced mGluR-LTD

Because *Akt1* and *Akt3* are the more similarly expressed isoforms in the hippocampus, we hypothesized that AKT1 and AKT3 may substitute for each other in regulating mGluR-LTD expression at

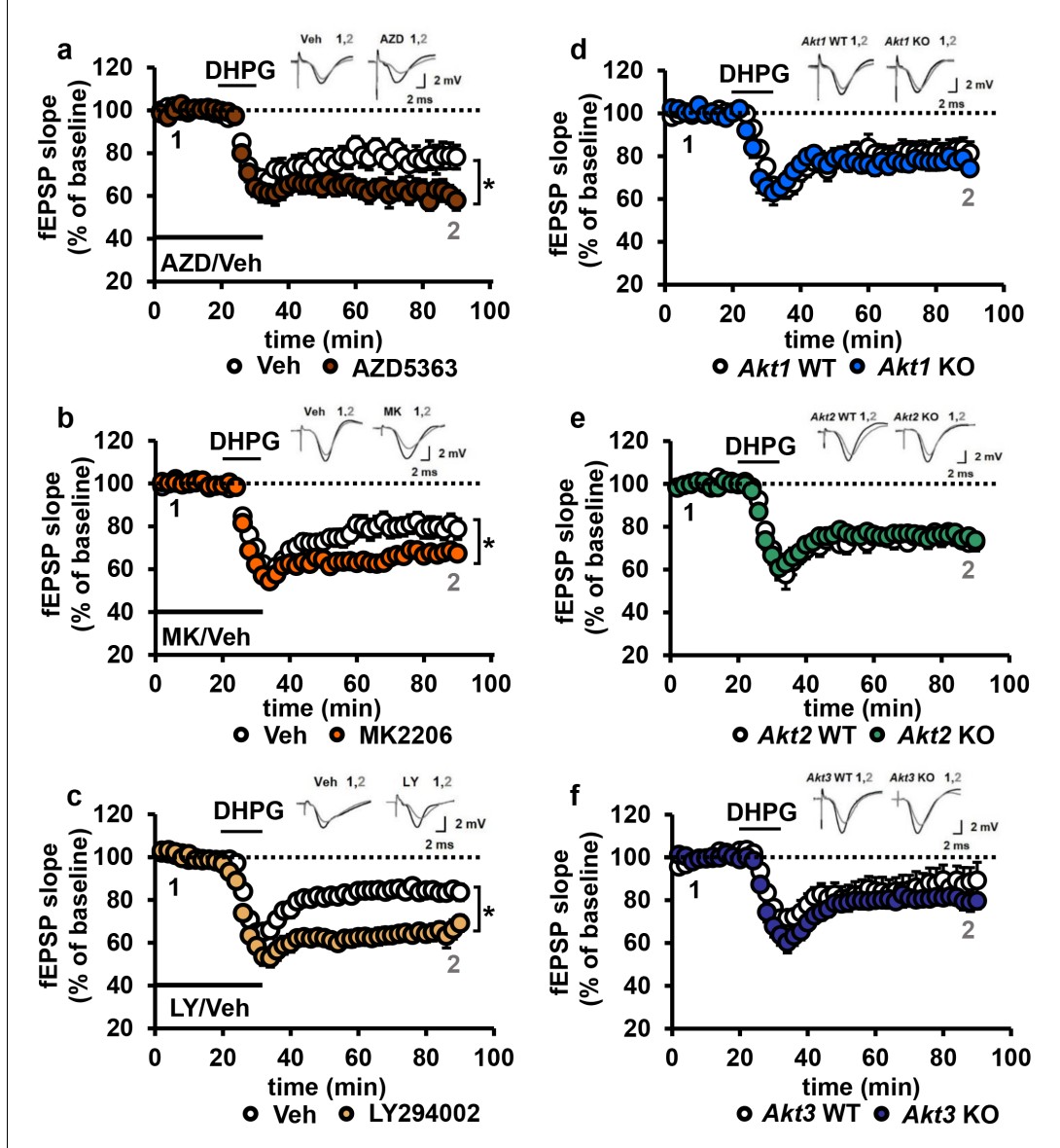

**Figure 7.** Blocking AKT activity leads to enhanced mGluR-LTD. (**a,b**) Blocking AKT activity using AZD5363 or MK2206 resulted in enhanced mGluR-LTD induced with 3,5 RS-DHPG (100 µM) in hippocampal area CA1 (AZD $F_{(1,23)}$=5.010, p=0.035; MK $F_{(1,40)}$=7.216, p=0.010), n = 12–20 slices/group, 5–6 mice/group. (**c**) Blocking PI3K using LY294002 resulted in enhanced mGluR-LTD ($F_{(1,27)}$=10.809, p=0.003), n = 13–16 slices/group, 7–8 mice/group. (**d–f**) *Akt1*, *Akt2* or *Akt3* KO mice show normal mGluR-LTD compared to WT controls (p>0.05), n = 13–20 slices/group, 5–7 mice/group. *p<0.05.
DOI: https://doi.org/10.7554/eLife.30640.014

CA3-CA1 synapses. Therefore, combined *Akt1* and *Akt3* deletion may result in enhanced mGluR-LTD, reproducing the effect of pan-AKT inhibition with either AZD or MK. Because *Akt1*/*Akt3* double KO mice are embryonic lethal (*Yang et al., 2005*), we used a Cre-mediated strategy of selective *Akt1* disruption by crossing mice with floxed alleles of *Akt1* to a forebrain-specific neuronal Cre recombinase mouse line (T29) in an *Akt3* KO background (cA1F/A3K mice) (*Figure 8a*). Using this strategy, AKT3 is removed and AKT1 levels are significantly reduced in the hippocampus of cA1F/A3K mice compared with WT mice (*Figure 8b,c*). Immunostaining of the hippocampus showed *Akt1* was mostly deleted from excitatory pyramidal neurons in area CA1, while other neurons, most likely inhibitory neurons, still expressed AKT1 (*Figure 8d*). Next, we examined mGluR-LTD in cA1F/A3K mice. Consistent with our results from the AKT inhibitor experiments, we found that mGluR-LTD was enhanced in cA1F/A3K mice compared with WT mice (*Figure 8e*). Thus, concomitant *Akt1* and *Akt3*

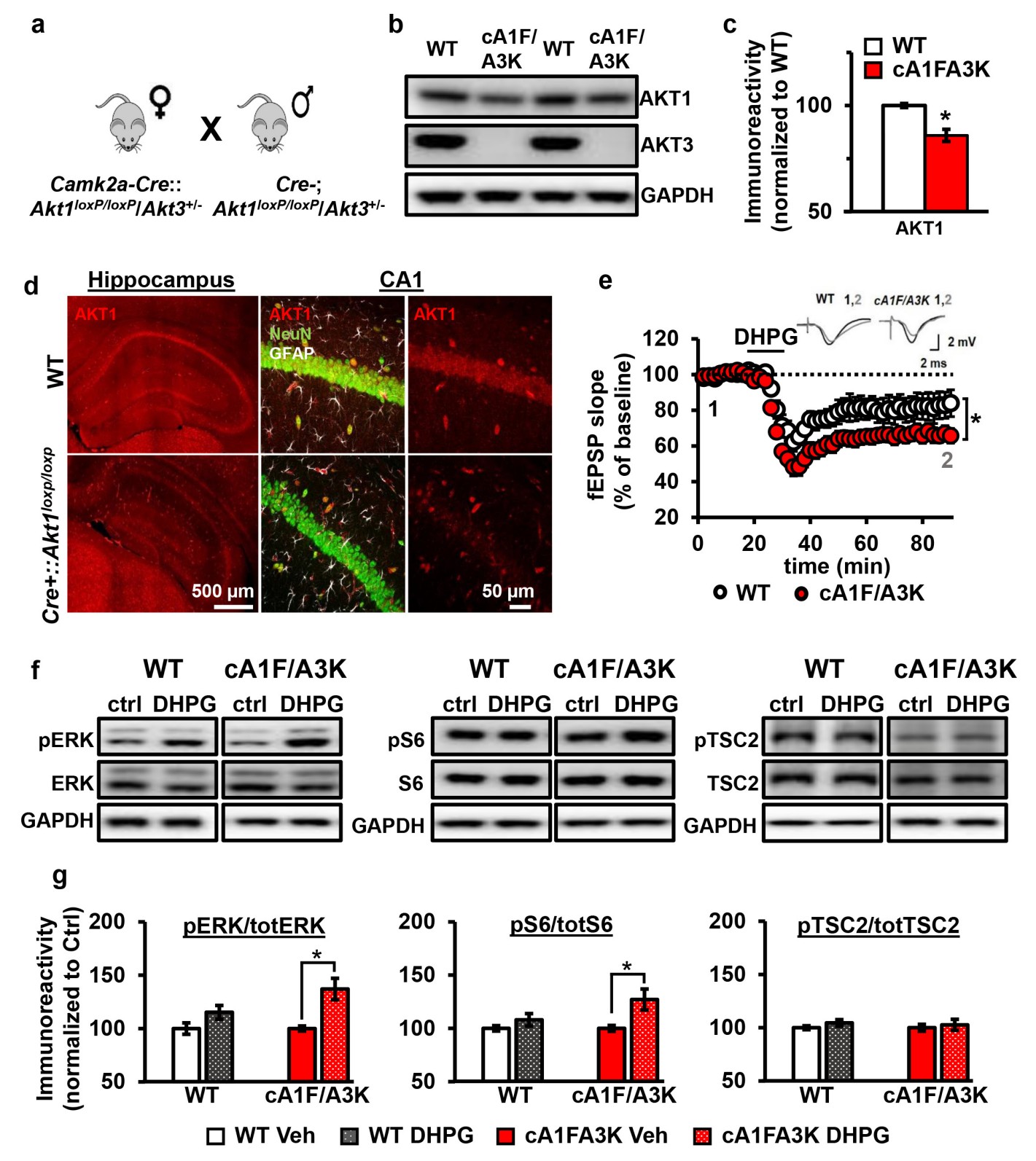

**Figure 8.** AKT1 and AKT3 are involved in mGluR-LTD. (**a**) Double *Akt1* and *Akt3* mutant (cA1F/A3K) mice with Cre-mediated removal of both *Akt1* alleles in the *Akt3* KO background were generated with WT littermate controls by breeding *Camk2a-Cre::Akt1^{loxP/+}/Akt3^{+/-}* female mice with *Akt1^{loxP/+}/Akt3^{+/-}* males. (**b,c**) Western blot analysis showing significantly reduced AKT1 levels and no AKT3 expression in the hippocampus of cA1F/A3K mice compared with WT mice (AKT1 levels: $t_{(6)}$=3.802, p=0.008). (**d**) Immunostaining confirming reduced neuronal AKT1 expression by *Camk2a*-driven Cre
*Figure 8 continued on next page*

*Figure 8 continued*

removal of *Akt1* in the hippocampus, especially in the pyramidal cell body layer of CA1. (e) mGluR-LTD is enhanced in cA1F/A3K hippocampal slices ($F_{(1,30)}$=7.923, p=0.009), n = 16 slices/group, 5 mice/group. (f,g) Western blot analysis showing significantly increased phosphorylation of ERK Thr202/Tyr204 and S6 S235/236 but not TSC2 S939 in cA1F/A3K hippocampal slices at 60 min post-DHPG, while WT slices had no differences (pERK: WT Ctrl vs. DHPG $t_{(22)}$=1.814, p=0.083, cA1F/A3K Ctrl vs. DHPG $t_{(14)}$=2.918, p=0.011; pS6: WT Ctrl vs. DHPG $t_{(22)}$=1.252, p=0.224, cA1F/A3K Ctrl vs. DHPG $t_{(14)}$=2.647, p=0.019; pTSC2: WT Ctrl vs. DHPG $t_{(20)}$=1.252, p=0.225, cA1F/A3K Ctrl vs. DHPG $t_{(14)}$=0.374, p=0.714). n= 8–12 slices/group, 4–6 mice/group, *p<0.05.

DOI: https://doi.org/10.7554/eLife.30640.015

The following figure supplement is available for figure 8:

**Figure supplement 1.** cA1FA3K mutant mice show decreased phosphorylation of TSC2 S939 under basal conditions (pTSC2: $t_{(19)}$=4.401, p=0.0004; 4–6 mice/group, *p<0.05.

DOI: https://doi.org/10.7554/eLife.30640.016

deletion results in enhanced mGluR-LTD, supporting the idea that the AKT1 and AKT3 isoforms function in hippocampal mGluR-LTD and can compensate for each other.

To investigate the signaling pathways underlying this mGluR-LTD enhancement, we next examined the activity of several proteins known to be involved in mGluR-LTD: extracellular regulated kinase (ERK) (*Gallagher et al., 2004*), ribosomal protein S6 (*Antion et al., 2008a*), and TSC2 (*Auerbach et al., 2011*). At 60 min post-DHPG, when cA1F/A3K slices show enhanced mGluR-LTD, we found that phosphorylation levels of ERK and S6 were not different in DHPG-treated WT slices but significantly increased in DHPG-treated cA1F/A3K slices compared with vehicle treatment. In contrast, phosphorylation of TSC2 showed no difference in either WT or cA1F/A3K slices 60 min post-DHPG compared to vehicle treatment. Interestingly, however, basal levels of TSC2 phosphorylation were significantly reduced in cA1FA3K slices compared with WT slices (*Figure 8—figure supplement 1*). Together, these results show that removal of both AKT1 and AKT3 isoforms leads to altered ERK and S6 activity regulation, which then most likely leads to enhanced synaptic depression upon mGluR1 stimulation.

## Discussion

Our studies discovered novel distinct roles for the different *Akt* isoforms in the brain. We show, for the first time, that AKT2 expression is localized to astrocytes in the hippocampus, while AKT1 and AKT3 are primarily but differentially expressed in neurons. We also show for the first time that these differences in isoform expression are associated with different functions in the brain involving synaptic plasticity. We demonstrate that the AKT1 isoform is critical for the expression of L-LTP and regulating activity-induced protein synthesis that supports L-LTP in area CA1. Interestingly, in contrast to the apparent singular requirement for AKT1 in L-LTP, we found that AKT1 and AKT3 serve overlapping roles to inhibit the expression of mGluR-LTD in area CA1. To our knowledge, this is the first study to identify AKT isoform-specific expression in the hippocampus and AKT isoform-specific activity underlying different forms of hippocampal synaptic plasticity.

Because AKT controls numerous cellular processes, uncovering isoform differences is important to improve understanding about normal AKT function. Moreover, isoform-specific roles for AKT have been associated with different disease pathologies. The AKT2 isoform, for example, has been implicated in diabetes. A missense mutation in *AKT2* that causes loss of AKT2 function leading to insulin resistance was identified in a family with diabetes (*George et al., 2004*). Similarly, AKT2-deficient mice have resistance to insulin and a subset develop diabetes-mellitus-like syndrome (*Cho et al., 2001a*). AKT2 also has been associated with glioma, a type of brain cancer with malignant tumors derived from glial cells. High-grade gliomas have been shown to overexpress AKT2 (*Zhang et al., 2010*). Although this correlation suggests a role for AKT2 in astrocytic function, we were surprised that our immunostaining in the hippocampus showed AKT2 to be present predominantly in astrocytes. Cre recombinase under the nestin promoter in neural progenitor cells, which generate both neurons and astrocytes, completely ablated AKT2 in the hippocampus (*Figure 2—figure supplement 1c*), whereas neuron-restricted Cre recombinase had no effect on hippocampal AKT2 levels (*Figure 2—figure supplement 1a*). Deducing from these results suggests that AKT2 is only expressed in astrocytes. However, it may be that our strategies have limited sensitivity to detect

trace amounts of AKT2 in neurons. Expression data from other studies also do not provide conclusive answers to where AKT2 is expressed in the brain. For example, the Allen Brain Atlas shows *Akt2* mRNA expression to be merely neuronal, whereas single-cell RNA sequencing of hippocampal cells identified *Akt2* only in astrocytes (*Zeisel et al., 2015*). In a proteome analysis of the brain, expression of all three AKT isoforms were reported in all brain cell types in almost equal amounts (*Sharma et al., 2015*). Although these methods generate large amounts of data and are important to examine the transcriptome on a large scale, these studies have not specifically verified AKT isoform expression in more depth. Close examination of our immunostaining revealed little or no detectable AKT1 and AKT3 expression in astrocytes (*Figure 2*), although we cannot exclude that they may be present. Therefore, it may be possible that AKT2 expression is restricted to hippocampal astrocytes or becomes restricted post-developmentally or that signals from other brain cell types in vivo restrict AKT2 expression to astrocytes. Although we found no evidence of a role for AKT2 in the synaptic plasticity paradigms tested in this study, AKT2 might play a role in synaptic plasticity as it relates to astrocytic function for supporting neuronal activity (*Papouin et al., 2017*).

The AKT1 isoform specifically has been linked to schizophrenia, a neurological disorder believed to represent brain dysconnectivity with a molecular basis in aberrant synaptic plasticity (*Stephan et al., 2006*). AKT1 is one of the effectors in the well-established pathway from neuregulin 1 (*NRG1-ERBB4-PI3K-AKT1* pathway), where each of these components have genetic polymorphisms that have been linked to schizophrenia (*Emamian et al., 2004*; *Hatzimanolis et al., 2013*). These polymorphisms may lead to lower AKT1 protein levels in schizophrenia (*Emamian et al., 2004*; *van Beveren et al., 2012*), although increased AKT1 expression and activity also have been reported in schizophrenia (*Hino et al., 2016*; *Kumarasinghe et al., 2013*). Human olfactory neurosphere-derived cells collected from schizophrenia patients were shown to have reduced protein synthesis (*English et al., 2015*). Additionally, the antipsychotic haloperidol used to treat schizophrenia can activate the AKT pathway, thereby inducing protein synthesis (*Bowling et al., 2014*). Together, these reports may suggest protein synthesis is reduced in schizophrenia, which may be due to reduced AKT1 signaling. Because protein synthesis is crucial to maintain L-LTP (*Sweatt, 2016*), our finding that *Akt1* deletion results in the impairment of L-LTP (*Figure 4*) and the associated protein synthesis response (*Figure 5*) suggests AKT1 may be an important factor in the synaptic and cognitive deficits observed in schizophrenia (*Stephan et al., 2006*).

The AKT3 isoform is expressed in a highly overlapping pattern with AKT1 in the hippocampus (*Figure 1*), suggesting overlapping roles with AKT1, but there are also notable differences between their expression. For example, AKT3 shows expression in the neuropil (*Figures 1* and *2*). AKT3 is known to control brain size (*Easton et al., 2005*; *Tschopp et al., 2005*). Mutations that result in enhanced AKT3 activity lead to increased brain growth (*Lee et al., 2012*), while deletion of *AKT3* is involved in microcephaly (*Gai et al., 2015*). AKT3 may control brain growth through regulation of protein synthesis. Deletion of *Akt3* is known to reduce phosphorylation of the ribosomal protein S6 (*Easton et al., 2005*), which is an effector in the AKT-mTOR pathway leading to protein synthesis. Intriguingly, our results show that only AKT1-deficient mice display impaired protein synthesis-dependent L-LTP, while deletion of *Akt3* has no effect (*Figure 4*). These data are consistent with the idea that while both AKT1 and AKT3 can regulate neuronal protein synthesis, AKT1 is specifically recruited to support translation involved in L-LTP.

What mechanisms might underlie the differential regulation of protein synthesis by AKT1 and AKT3? Phosphorylation of S6 on S235/236 is increased after L-LTP, which correlates with increased translation (*Antion et al., 2008b*). Consistent with our finding that *Akt1* deletion leads to an impaired protein synthesis response, *Akt1* KO slices fail to increase S6 phosphorylation following stimulation, whereas *Akt3* KO slices still display dynamic range in protein synthesis and S6 activation (*Figure 5*). Noteworthy, previously it was found that basal phosphorylation of S6 is reduced in *Akt3* KO mice but not in *Akt1* KO mice (*Easton et al., 2005*), which we were able to confirm (*Figure 5e, f*). These results show distinct roles for AKT1- and AKT3-mediated protein synthesis. AKT1 is the activity-regulated isoform, converting external stimuli into protein synthesis, while AKT3 may be more important for steady-state protein synthesis. This distinction may also correlate with the regional differences we observed in hippocampal AKT1 and AKT3 expression (*Figures 1* and *2*). Thus, in regions where AKT3 expression predominates like area CA3, *Akt3* KO slices may show altered activity-induced protein synthesis, following stimulation of the mossy fiber-CA3 circuit for

instance. Additionally, future experiments aimed at finer localization of AKT1 and AKT3 may help to answer this question.

mGluR-LTD has been widely studied in relation to the neurodevelopmental disorder fragile X syndrome (FXS) and shown to be enhanced in hippocampal slices from *Fmr1* KO mice, a mouse model of FXS (*Huber et al., 2002*). mGluR-LTD requires protein synthesis, leading to AMPA receptor internalization (*Huber et al., 2001*). AKT has been implicated in the cascade mediating this protein synthesis after mGluR stimulation (*Hou and Klann, 2004*). Studies inhibiting upstream and downstream signaling of AKT have suggested that mGluR stimulation activates a cascade through PI3K-AKT-TSC-mTORC1-S6K to protein synthesis. However, there are conflicting findings about the role of this signaling cascade in mGluR-LTD. In support of the idea that AKT functions in this pathway to facilitate mGluR-LTD, pharmacological studies inhibiting PI3K (*Hou and Klann, 2004*; *Potter et al., 2013*) or mTORC1 (*Potter et al., 2013*; *Sharma et al., 2010*) have reported mGluR-LTD impairment. On the other hand, inhibiting mTORC1 also has been reported to have no effect on mGluR-LTD (*Auerbach et al., 2011*; *Potter et al., 2013*). Likewise, loss of S6K1 has been shown to have no effect on mGluR-LTD, while S6K2 loss results in enhanced mGluR-LTD (*Antion et al., 2008a*). Some of these divergent results may be due to methodological differences or the developmental stage at which experiments were performed. However, it remains that none of these earlier studies directly examined the role of AKT activity in mGluR-LTD. To address this gap, we targeted AKT directly using multiple approaches, including pharmacological agents and genetic removal of *Akt*.

Our results strongly support the idea that AKT normally acts to inhibit mGluR-LTD. We found that blockade of AKT activity with multiple separate approaches, using two different AKT inhibitors, genetic deletion of both *Akt1* and *Akt3* in CA1 as well as PI3K inhibition, all resulted in enhanced mGluR-LTD (*Figures 7* and *8*). By using two AKT inhibitors that have different mechanisms of action on AKT, complemented by genetically removing AKT, we addressed potential off-target effects of the compounds and provide converging evidence that the enhanced mGluR-LTD is due to specific inhibition of AKT. Developmental effects of AKT removal also are unlikely to contribute to the observed LTD enhancement as single *Akt* isoform deletion did not affect mGluR-LTD (*Figure 7*). Additionally, in the double *Akt1/Akt3* deletion experiment, AKT1 is removed postnatally, eliminating potential confounds from the loss of multiple AKT isoforms during development. Combined, these data show that the role of AKT in mGluR-LTD may be more complex than originally thought. AKT may not simply be a facilitator of general mTORC1-mediated translation but may only be involved in regulating distinct pools of mGluR-induced protein synthesis, some of which may act to inhibit mGluR-LTD. In agreement with this idea, the AKT-TSC pathway has been suggested to act as a brake on mTOR-regulated protein synthesis, while the ERK pathway, which is also activated by mGluR stimulation, promotes protein synthesis (*Auerbach et al., 2011*). Indeed, at 60 min post-DHPG, we found increased phosphorylation levels of ERK and S6 in cA1F/A3K mice compared with vehicle treatment but not in WT mice. This suggests that AKT1 and AKT3 removal may relieve the brake on protein synthesis after mGluR stimulation, leading to altered protein synthesis or AMPA receptor internalization and subsequently enhanced mGluR-LTD. Future studies examining these possibilities will be important to resolve the role of AKT in regulating these important pathways in mGluR-LTD.

In summary, this study provides valuable new insights into the role of AKT and its isoforms in the brain. The results show expression differences between AKT isoforms in the hippocampus, affecting their role in synaptic plasticity and protein synthesis. Hence, compounds targeting single AKT isoforms may be an attractive therapeutic target for treating disorders and diseases that impact synaptic plasticity.

## Materials and methods

**Key resources table**

| Reagent type (species) or resource | Designation | Source or reference | Identifiers | Additional information |
|---|---|---|---|---|
| Antibody | Rabbit anti-AKT1 (Western blot) | Cell Signaling | Cat# 2938; RRID:AB_915788 | 1:1000 |

*Continued on next page*

*Continued*

| Reagent type (species) or resource | Designation | Source or reference | Identifiers | Additional information |
|---|---|---|---|---|
| Antibody | Rabbit anti-AKT1 (Immunostaining) | Cell Signaling | Cat# 75692; RRID:AB_2716309 | 1:100 |
| Antibody | Rabbit anti-AKT1 phospho serine 473 | Cell Signaling | Cat# 9081; RRID:AB_11178946 | 1:1000 |
| Antibody | Mouse anti-AKT2 (Western blot) | LSBio | Cat# LS-C156232; RRID:AB_2716310 | 1:1000 |
| Antibody | Rabbit anti-AKT2 (Immunostaining) | Cell Signaling | Cat# 2964; RRID:AB_331162 | 1:100 |
| Antibody | Rabbit anti-AKT2 phospho serine 474 | Cell Signaling | Cat# 8599; RRID:AB_2630347 | 1:1000 |
| Antibody | Rabbit anti-AKT3 (Western blot) | Cell Signaling | Cat# 14293; RRID:AB_2629491 | 1:1000 |
| Antibody | Mouse anti-AKT3 (Western blot) | Cell Signaling | Cat# 8018; RRID:AB_10859371 | 1:1000 |
| Antibody | Rabbit anti-AKT3 (Immunostaining and immunoprecipitation) | Cell Signaling | Cat# 14982; RRID:AB_2716311 | 1:100 Immunostaining 1:50 immunoprecipitation |
| Antibody | Rabbit anti-AKT phospho serine 473 | Cell Signaling | Cat# 3787; RRID:AB_331170 | 1:1000 |
| Antibody | Rabbit anti-AKT | Cell Signaling | Cat# 4685; RRID:AB_2225340 | 1:2000 |
| Antibody | Rabbit anti-TSC2 S939 phospho serine 939 | Cell Signaling | Cat# 3615; RRID:AB_2207796 | 1:1000 |
| Antibody | Rabbit anti-TSC2 | Cell Signaling | Cat# 3635; RRID:AB_10692893 | 1:1000 |
| Antibody | Rabbit anti-PDK1 phospho serine 241 | Cell Signaling | Cat# 3061; RRID:AB_2161919 | 1:3000 |
| Antibody | Rabbit anti-PDK1 | Cell Signaling | Cat# 3062; RRID:AB_2236832 | 1:2000 |
| Antibody | Rabbit anti-GAPDH | Cell Signaling | Cat# 2118; RRID:AB_561053 | 1:10000 |
| Antibody | Mouse anti-$\beta$ actin | Abcam | Cat# 8226; RRID:AB_306371 | 1:10000 |
| Antibody | Mouse anti-NeuN | Novus | Cat# NBP1-92693; RRID:AB_11036146. | 1:1000 |
| Antibody | Chicken anti-GFAP | PhosphoSolutions | Cat# 621-GFAP; RRID:AB_2492125 | 1:1000 |
| Antibody | Rabbit anti-GSK-3$\beta$ phospho serine 9 | Cell Signaling | Cat# 9323; RRID:AB_2115201 | 1:3000 |
| Antibody | Rabbit anti- GSK-3$\beta$ | Cell Signaling | Cat# 9315; RRID:AB_490890 | 1:3000 |
| Antibody | Mouse anti-GluA2 | NeuroMAB | Clone N355/1; RRID:AB_2315839 | 1:1000 |
| Antibody | Rabbit anti-S6 phospho S235/236 | Cell Signaling | Cat# 4856; RRID:AB_2181037 | 1:2000 |
| Antibody | Rabbit anti-S6 | Cell Signaling | Cat# 2217; RRID:AB_331355 | 1:2000 |
| Antibody | Mouse anti-puromycin | Abcam | 12D10; RRID:AB_2566826 | 1:1000 |
| Antibody | Rabbit anti-phospho-p44/42 MAPK (Erk1/2) (Thr202/Tyr204) | Cell Signaling | Cat# 9101; RRID:AB_331646 | 1:4000 |
| Antibody | Rabbit p44/42 MAPK (Erk1/2) Antibody | Cell Signaling | Cat#9102; 9102S RRID:AB_330744 | 1:4000 |

*Continued on next page*

*Continued*

| Reagent type (species) or resource | Designation | Source or reference | Identifiers | Additional information |
|---|---|---|---|---|
| Antibody | Donkey anti-Rabbit-Cy3 | Jackson ImmunoResearch | Cat# 711-165-152; RRID:AB_2307443 | 1:200 |
| Antibody | Donkey anti-Mouse Alexa 488 | Jackson ImmunoResearch | Cat# 715-545-150; RRID:AB_2340846 | 1:200 |
| Antibody | Donkey anti-Chicken Alexa 647 | Jackson ImmunoResearch | Cat# 703-605-155; RRID:AB_2340379 | 1:200 |
| Chemical compound, drug | Hoechst | Sigma | Cat# H6024 | 1:3000 |
| Antibody | Goat anti-mouse HRP | Promega | W4020 | 1:5000 |
| Antibody | Goat anti-rabbit HRP | Promega | W4011 | 1:5000 |
| Chemical compound, drug | MK-2206-2HCl | Selleckchem | Cat # S1078 | |
| Chemical compound, drug | AZD5363 | Selleckchem | Cat # S1078 | |
| Chemical compound, drug | LY294002 | Sigma | Cat # L9908 | |
| Other | A/G agarose beads | Pierce | Cat# 20421 | |
| Chemical compound, drug | Protease inhibitor | Sigma | P8340 | |
| Chemical compound, drug | Phosphatase inhibitor II | Sigma | P5726 | |
| Chemical compound, drug | Phosphatase inhibitor III | Sigma | P0044 | |
| Chemical compound, drug | Puromycin | Sigma | P8833 | |
| Chemical compound, drug | (RS)−3,5- DHPG | Tocris | Cat# 0342 | |
| Commercial assay or kit | ECL Prime Western blot Detection | GE Healthcare | RPN2236 | |
| *Mus Musculus, Akt1$^{tm1Mbb}$, C57BL/6 ()* | *Akt1* KO | Jackson Laboratory | Stock # 004912; RRID:IMSR_JAX:004912 | |
| *Mus Musculus, Akt2$^{tm1.1Mbb}$, C57BL/6 ()* | *Akt2* KO | Jackson Laboratory | Stock # 006966; RRID:IMSR_JAX:006966 | |
| *Mus Musculus, Akt3$^{tm1.3Mbb}$, C57BL/6 ()* | *Akt3* KO | *Easton et al. (2005)*; PMCID: PMC549378 | | Obtained from Birnbaum lab (UPenn) in 2012 |
| *Mus musculus, Akt1$^{tm2.2Mbb}$, C57BL/6 ()* | *Akt1$^{loxP/loxP}$* | Jackson Laboratory | Stock #026474; RRID:IMSR_JAX:026474 | |
| *Mus Musculus, Akt2$^{tm1.2Mbb}$, C57BL/6 ()* | *Akt2$^{loxP/loxP}$* | Jackson Laboratory | Stock #026475; RRID:IMSR_JAX:026475 | |
| *Mus Musculus, Tg$^{(CamkIIa-Cre)T29Stl}$, C57BL/6 ()* | *CamK2a-Cre* | *Hoeffer et al., 2008*; PMCID: PMC2630531 | | Obtained from Kelleher lab (MIT) in 2008 |
| *Mus Musculus, Tg$^{(Eno2-cre)39Jme}$, C57BL/6 ()* | *Eno2*-Cre -or- *NSE39*-Cre | Jackson Laboratory | Stock # 005938; RRID:IMSR_JAX:005938 | |
| *Mus Musculus, Tg$^{(Nes-cre)1Kln}$, C57BL/6 ()* | *Nestin*-Cre | Jackson Laboratory | Stock # 003771; RRID:IMSR_JAX:003771 | |
| Software, algorithm | IBM SPSS Statistics | IBM Analytics | RRID:SCR_002865 | |
| Software, algorithm | pCLAMP software | Molecular Devices | RRID:SCR_011323 | |
| Software, algorithm | ImageQuant TL | GE Healthcare | RRID:SCR_014246 | |
| Software, algorithm | ICY Imaging | Other | RRID:SCR_010587 | Open Source; http://icy.bioimageanalysis.org/ |
| Other | Cryostat | Leica CM1850 | | |
| Other | Nikon A1R Laser Scanning Confocal | Nikon | | |
| Other | Vibratome | Leica VT1200S | | |
| Other | Brain slice incubation chamber | Scientific Systems Design Inc | BSC1 | |
| Other | Proportional Temperature Controller | Scientific Systems Design Inc | PTC03 | |

*Continued on next page*

Continued

| Reagent type (species) or resource | Designation | Source or reference | Identifiers | Additional information |
|---|---|---|---|---|
| Other | Microelectrode AC Amplifier Model 1800 | A-M Systems | Cat # 70000 | |
| Other | Axon CNS Digidata 1440A | Molecular Devices | | |
| Other | Sonicator XL2000 | QSonica | | |
| Other | Synergy 2 Multimode reader | BioTek | | |
| Other | XCell II Surelock Western blot | ThermoFisher Scientific | EI0002 | |
| Other | Novex 4–12% Bis-Tris gels (15 or 26 well) | Life Technologies | 15 well: NP0336 | |
| | | | 26 well: WG1403 | |
| Other | FluorChem E system | Proteinsimple | | |

## Mice

*Akt1* KO (*Cho et al., 2001b*), *Akt2* KO (*Cho et al., 2001a*) and *Akt3* KO (*Easton et al., 2005*) male mice were generated in the C57/BL6 background. To generate mice that have conditional *Akt1* or *Akt2* removal in excitatory neurons in the forebrain, we bred *Akt1*$^{loxP/loxP}$(*Wan et al., 2012*) or *Akt2-*$^{loxP/loxP}$ (*Leavens et al., 2009*) with *Camk2a*-Cre (T-29–1) (*Wong et al., 2015*), rat neuron-specific enolase, NSE or *Eno2*-Cre (NSE39-Cre, Jax Labs) or *Nestin*-Cre (Jax Labs) mice, all in the C57/BL6 background. Age-matched wild-type (WT) male littermates were used as controls for this study. Mice were housed at 21°C, maintained on a 12 hr light/dark schedule with food and water available *ad libitum* and tested at 5–20 weeks of age, depending on the experimental protocol. All procedures were approved by the University of Colorado, Boulder Animal Care and Use Committee.

## Immunohistology

WT and *Akt* mutant mice were anesthetized using a mixture of pentobarbital sodium and phenytoin sodium (Euthanasia III). When they were unresponsive to a toe pinch, mice were intracardially perfused with PBS followed by 4% paraformaldehyde (PFA). After perfusion, brains were isolated and kept in 4% PFA for 24 hr at 4°C. Brains were then transferred to 30% sucrose in PBS for at least 24 hr at 4°C. Next, brains were sectioned into 30 μm coronal slices on a cryostat (Leica) and stored at −20°C in cryoprotectant (20% glycerol/2% DMSO in phosphate buffer) until used for immunohistology. Fluorescent immunostaining was performed as described previously with minor changes (*Levenga et al., 2013*). Briefly, free-floating brain sections were washed with PBS and submitted to heat-mediated antigen retrieval in citrate buffer (10 mM, pH 6) if needed. Slices were blocked for 1 hr at room temperature in staining buffer containing 0.05 M Tris pH 7.4, 0.9% NaCl, 0.25% gelatin, 0.5% TritonX-100, and 2% donkey serum. Slices were then incubated overnight at 4°C with primary antibodies against AKT1 (1:100, Cell Signaling D9R8K, central amino acid of epitope is D108), AKT2 (1:100, Cell Signaling 5B5, central amino acid of epitope is P471), AKT3 (1:100, Cell Signaling E1Z3W, central amino acid of epitope is E131), NeuN (1:1000, Millipore Mab377), and GFAP (1:1000, PhosphoSolutions 621-GFAP) diluted in staining buffer with 2% donkey serum. After three washes in PBS, the brain slices were incubated at room temperature for 2 hr in Cy3-conjugated anti-rabbit, 488-conjugated anti-mouse and Cy5-conjugated anti-chicken secondary antibodies (1:200, Jackson Immunoresearch) diluted in staining buffer with Hoechst dye (1:3000). Following three washes in PBS, slices were mounted and coverslipped with Mowiol. Brain slices were imaged using the Nikon A1R confocal microscope. Z-stacks through the entire thickness of the slice were taken at 20x and 100x. All microscope parameters were held constant across all slices.

## Field electrophysiology

Long-term potentiation (LTP) and long-term depression (LTD) in the CA3-CA1 circuit from acute 400 μm transverse hippocampal slices were recorded as previously described (*Wong et al., 2015*; *Levenga et al., 2013*). Hippocampal slices were derived from mice 5–6 weeks of age for LTD experiments and greater than 10 weeks of age for LTP experiments. Briefly, slices were maintained in an interface chamber at 32°C infused with oxygenated ACSF containing (in mM): 125 NaCl, 2.5 KCl,

1.25 a$H_2PO_4$, 25 $NaHCO_3$, 25 D-glucose, 2 $CaCl_2$ and 1 $MgCl_2$. For LFS-LTD recordings, slices were maintained in ACSF containing 2.5 mM $CaCl_2$. Slices recovered in the chamber at least 60 min prior to recordings. Next, constant-current stimuli (100 μs) were delivered with a bipolar silver electrode placed in the stratum radiatum of area CA3, and field excitatory postsynaptic potentials (fEPSPs) were recorded with an electrode filled with ACSF in the stratum radiatum of area CA1. fEPSPs were monitored by delivering stimuli at 0.033 Hz and measuring their slopes using pCLAMP10 (Molecular Devices). Before LTP or LTD induction, a stable baseline was established for 20–30 min with a stimulus intensity of 40–50% of the maximum fEPSP. Late-phase LTP (L-LTP) was induced by four trains of 100 Hz high-frequency stimulation (HFS) for 1 s with a 5-min intertrain interval. LFS-LTD was induced by 900 stimuli of 1 Hz. For drug treatments, 100 μM (RS)−3,5-DHPG (Tocris Bioscience, Minneapolis, MN), 30 μM MK-2206 2HCL (Selleckchem, Houston, TX), 30 μM AZD5363 (Selleckchem), 50 μM LY294002 (Sigma, St. Louis, MO), or vehicle was applied to the ACSF.

## Puromcyin-labeled protein synthesis assay

Newly synthesized proteins were labeled using an adaption of the SUnSET protocol as described in (*Hoeffer et al., 2013*). To control for metabolic state differences between mice, only slices generated from a single mouse are used for all testing conditions. Briefly, hippocampal slices were prepared for field electrophysiology as described earlier. Immediately after HFS was delivered to induce L-LTP, hippocampal slices were incubated with puromycin (10 μg/ml) in ACSF for 30 min at 32°C. After puromycin labeling, slices were flash frozen on dry ice and processed for Western blot analysis.

## Drug concentration assay

Hippocampal slices were prepared as described above but were maintained at room temperature for 1 hr in oxygenated ACSF prior to being transferred to vials containing oxygenated ACSF at 32°C for 1 hr. Then, slices were treated with AKT inhibitors for 30 min and flash frozen on dry ice to be processed for Western blot analysis.

## Immunoprecipitation

Because a specific phospho-AKT3 antibody was not available to measure phosphorylation levels of AKT3, AKT3 protein was purified from AKT inhibitor-treated slices and from *Akt1* KO and WT hippocampal tissue. These tissues were homogenized by sonication in ice-cold immunoprecipitation (IP) buffer containing in mM: 40 HEPES pH 7.5, 150 NaCl, 10 pyrophosphate, 10 glycerophosphate, 1 orthovanadate, 1 EDTA, 1 EGTA, and 50 NaF; 1X protease inhibitor cocktail III and phosphatase inhibitor cocktails II and III (Sigma); and 0.1% Triton X-100. 100 μg of the lysates were diluted to 100 μL in IP buffer and incubated with anti-AKT3 antibody (Cell Signaling, Beverly, MA, E1Z3W) at 100:1 v/v, shaking gently overnight at 4°C. Samples were then incubated with 50 μL of slurry containing IgG-bound agarose beads (Pierce, Waltham, MA), shaking gently overnight at 4°C, followed by centrifugation at 2500 *g* for 4 min at 4°C. The bead pellet containing immunoprecipitated AKT3 protein was washed twice with IP buffer and resuspended with Laemmli buffer to twice the pellet volume. 20 μL of the samples were then used for western blotting to examine phospho-AKT3 levels with the phospho-AKT S473 antibody.

## Western blot analysis

Either freshly extracted tissue or treated hippocampal slices were flash frozen on dry ice. Soluble protein extracts were prepared for western blotting using procedures adapted from previous studies (*Wong et al., 2015*). Briefly, WT, mutant or drug-treated hippocampal slices or tissue were homogenized by sonication in lysis buffer containing (in mM): 10 HEPES pH 7.4, 150 NaCl, 50 NaF, 1 EDTA, 1 EGTA, and 10 $Na_4P_2O_7$ with 1x protease inhibitor cocktail III (Sigma), 1x phosphatase inhibitor cocktails II and III (Sigma), 1% Triton-X, and 1% Igepal. 20- or 30 μg protein samples were prepared in Laemmli sample buffer and resolved using 4–12% Bis-Tris gradient gels. Proteins were then blotted on PDVF membrane and detected using standard techniques. Protein blots were blocked with 0.2% I-Block (Tropix-Thermo Fisher, Lafayette, CO) dissolved in Tris-buffered saline with 0.1% Tween-20 (TBS-T). Primary antibodies were diluted in I-Block and incubated for 24 hr at 4°C. Blots were washed in TBS-T followed by incubation with HRP-conjugated goat anti-rabbit or goat anti-

mouse secondary antibodies (1:5000, Promega, Madison, WI) to detect the primary antibody. Immunoreactive signals were visualized with enhanced chemiluminescence (GE Healthcare) and quantified in the linear range of detection as previously described (*Wong et al., 2015*). Signals were normalized by total protein levels for phospho-proteins or by loading control levels for total proteins. Primary antibodies used were puromycin (mouse monoclonal 12D10), AKT1 (1:1000, Cell Signaling C73H10), phospho-AKT1 S473 (1:1000, Cell Signaling D7F10), AKT2 (1:1000, LSBio, Seattle, WA, LS-C156232), phospho-AKT2 474 (1:1000, Cell Signaling D3H2), AKT3 (1:1000, Cell Signaling E2B6R), AKT3 (1:1000, Cell Signaling L47B1), phospho-AKT S473 (1:1000, Cell Signaling 736E11), pan-AKT (1:2000, Cell Signaling 11E7), phospho-GSK-3β S9 (1:3000, Cell Signaling 5B3), GSK-3β (1:3000, Cell Signaling 27C10), phospho-TSC2 S939 (1:1000, Cell signaling 3615), TSC2 (1:1000, Cell Signaling 3635), phospho-PDK1 S241 (1:3000, Cell Signaling 3061), PDK1 (1:2000, Cell Signaling 3062), phospho-S6 235/236 (1:2000, Cell Signaling 4856), S6 (1:2000, Cell Signaling 2217), phospho-ERK1/2 (1:4000, Cell Signaling 9101), ERK1/2 (1:4000, Cell Signaling 9102), Mouse anti-GluA2 (1:1000, NeuroMAB, Davis, CA), β-actin (1:10000, Abcam, Cambridge, MA), and GAPDH (1:10000, Cell Signaling 14C10).

## Statistical analysis

All data are presented as mean values ± SEM. Data were statistically evaluated using SPSS software. The use of parametric tests was determined with the Shapiro-Wilk test for normality and Student's *t* test or ANOVA were applied. Outliers were excluded using Grubb's method. Repeated measures (RM) ANOVA were used for electrophysiological experiments with time as the within-subjects factor. Significant effects were followed by Tukey's post hoc testing. All statistical tests were two-tailed with $p < 0.05$ considered as statistically significant.

## Acknowledgements

For technical assistance, resources, and funding, we thank Thomas F Franke, Michael Roche, CU Boulder BioFrontiers Microscopy Core, Alzheimer's Association MNIRGDP-12–258900 (CAH), NARSAD 21069 (CAH), NIH R01 NS086933 (CAH), Linda Crnic Institute Seed grant (CAH), NIH F31 NS083277 (HW), NIH T32 MH019524 (HW), Simons Foundation SFARI 27444 (CAH), and Sie Foundation (JL).

## Additional information

### Funding

| Funder | Grant reference number | Author |
|---|---|---|
| Sie Foundation | | Josien Levenga |
| National Institutes of Health | F31 NS083277 | Helen Wong |
| National Institutes of Health | T32 MH019524 | Helen Wong |
| Alzheimer's Association | MNIRGDP-12-258900 | Charles A Hoeffer |
| Simons Foundation | SFARI 27444 | Charles A Hoeffer |
| National Institutes of Health | R01 NS086933 | Charles A Hoeffer |
| Linda Crnic Institute for Down Syndrome | Seed Grant | Charles A Hoeffer |
| National Alliance for Research on Schizophrenia and Depression | 21069 | Charles A Hoeffer |

The funders had no role in study design, data collection and interpretation, or the decision to submit the work for publication.

## Author contributions
Josien Levenga, Ryan A Milstead, Conceptualization, Data curation, Formal analysis, Investigation, Methodology, Writing—original draft, Project administration, Writing—review and editing; Helen Wong, Conceptualization, Formal analysis, Writing—original draft, Writing—review and editing; Bailey N Keller, Conceptualization, Formal analysis, Investigation, Writing—original draft, Project administration, Writing—review and editing; Lauren E LaPlante, Data curation, Investigation; Charles A Hoeffer, Conceptualization, Resources, Formal analysis, Supervision, Funding acquisition, Investigation, Methodology, Project administration, Writing—review and editing

## Author ORCIDs
Josien Levenga (iD) http://orcid.org/0000-0002-9971-6337
Ryan A Milstead (iD) http://orcid.org/0000-0002-3333-853X
Charles A Hoeffer (iD) http://orcid.org/0000-0002-2036-0201

## Ethics
Animal experimentation: This study was performed in strict accordance with the recommendations in the Guide for the Care and Use of Laboratory Animals of the National Institutes of Health. All animals used in this study were handled according to the approved institutional animal care and use committee (IACUC) protocols (1311.02, 2541) of the University of Colorado-Boulder.

## Decision letter and Author response
Decision letter https://doi.org/10.7554/eLife.30640.020
Author response https://doi.org/10.7554/eLife.30640.021

# Additional files

## Supplementary files
• Transparent reporting form
DOI: https://doi.org/10.7554/eLife.30640.017

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
