## [Decision Letter]

Thank you for submitting your article "AKT isoforms have distinct hippocampal expression and roles in synaptic plasticity" for consideration by *eLife*. Your article has been reviewed by three peer reviewers, and the evaluation has been overseen by a Reviewing Editor and Richard Aldrich as the Senior Editor. The following individuals involved in review of your submission have agreed to reveal their identity: Kimberly M Huber (Reviewer #1); Clive Bramham (Reviewer #3).

The reviewers have discussed the reviews with one another and the Reviewing Editor has drafted this decision to help you prepare a revised submission. There were differing opinions about the suitability of the paper for the journal.

In this paper, the distribution and function of the AKT family of genes in hippocampus synaptic plasticity was studied. The authors find roles for AKT1 in L-LTP and redundant roles for AKT1 and AKT3 in the suppression of mGluR-LTD. The reviewers indicated the results to be original and show that AKT proteins may play distinct roles in the regulation of protein synthesis during LTD and LTP. However, several reservations were raised about the data to support this conclusion. Specifically, there were concerns raised regarding inconsistencies of the pharmacological findings and the results from AKT knockout mice. The collective concerns are delineated below.

Reviewer #1:

1) I commend the authors for doing a thorough job characterizing the AKT inhibitors in the slice preparation in Figure 3. However, there are some confusing results that may require more explanation. The MK inhibitor at 10um completely blocks P-AKT, but has no effect on its substrate GSK3B, whereas the AZD drug enhanced P-AKT, but inhibits GSK3B phosphorylation. Is AKT autophosphorylated? Or are the distinct effects of the inhibitors due to feedback onto PDK1? It would be good to see the same data in the AKT1 and AKT3 KO mice to see if similar or different effects are observed.

2) In Figure 4 pharmacological and genetic approach is used to study the effect of AKT on L-LTP. The Akt1 isoform is claimed to be the one involved in the maintenance of L-LTP and the target of the pharmacological blockers. Is somehow surprising that in AKT1 KO, the L-LTP seems to be only partially inhibited, whereas both of the pharmacological agents completely block this form of synaptic plasticity. Is the effect of the blockers specific? What is the effect of these inhibitors on the LTP induced in the AKT1 knockout mice.

3) In Figure 5, the authors show impaired protein synthesis in AKT1 KO after L-LTP induction. Are there any differences in protein synthesis rate under basal condition in each of the KO mice? That is not reported and would help to interpret the results.

4) In Figure 5, the effect of HFS on S6 phosphorylation in AKT1 and AKT3 ko is reported. The phosphorylation of S6 after HFS does not look different in the wild type slice, perhaps because they are saturated. What seems to be different is only the total amount of S6 that is reduced in the WT after HFS. It would be nice to see a loading control, actin or tubulin?, for these western. Also in this case, knowing the effect of AKT1 and AKT3 absence on basal level of S6 phosphorylation would help to interpret the results.

5) The most surprising, and therefore interesting, result in the paper, in my opinion, is Figure 7, where the authors show enhanced mGlu-LTD using the inhibitors of Akt, and this result is confirmed with PI3K inhibitors and in AKT1/AKT3 CA1 dKO. The authors discuss a role for AKT as a brake on mGlu-LTD and protein synthesis machinery induction. However, they present no data to support this conclusion. Because these authors used SunSET to examine effects of AKT deletion on HFS-induced protein synthesis, this should also be done with mGluR-LTD. This result may suggest that AKT plays distinct roles in protein synthesis regulation depending on the stimulus (LTP or LTD), which in my opinion is very novel and impactful. Also, examining the effects of the inhibitors and AKT deletion on DHPG-induced signaling to mTORC1 and other pathways would provide more mechanistic insight into how AKT does this. However, this could be beyond the scope of this present manuscript.

Reviewer #2:

The analysis of the mutant mice show that none of them affected synaptic plasticity up to an hour after any stimulation except the double mutant Akt1/3 in Figure 8 (mGluR-DHPG). Only L-LTP at 3 hours showed a decremental effect (incomplete LTP block) for Akt1 knockouts. They also find impaired protein synthesis in Akt1 knockouts and conclude that the effect on L-LTP is "through regulating protein synthesis". They have not shown a causal link, merely a correlation. These findings are a minimal advance, do not present any conceptual advances and are not of the highest technical standards.

The results with the drugs are problematic for the reasons mentioned. The result of enhanced mGluR-LTD with LY294002 is preliminary.

My conclusion is that this not a high interest paper worthy of publication in *eLife*.

Specific points in the order they arise in the manuscript:

1) The introduction to the Akt family is confusing and non-standard with respect to genes, gene families and proteins. The authors refer to "three highly related isoforms". There are many splice isoforms described for each of three genes (see Ensemble and UniProt). Nowhere to they explain that they are referring to three genes, and each one expresses multiple splice isoforms. A figure with relevant schematic sequence alignments of the proteins and explanation of the isoforms and paralogs would be useful, showing where the antibodies bind and where the location of domains that bind to the drugs. This would enable the reader to understand the limitations of the specificity of the reagents.

2) The differential expression of the three genes shown using immunolabelling in Figure 1 appears to describe differential expression, with Akt2 showing most differences to Akt1 and 3. An inspection of expression data in the Allen References Atlas shows Akt1,2 and 3 are expressed in all principal neurons, but Ak2 is at lower levels than either Akt1 and 3. Inspection of single-cell transcriptome data (Zeisel et al.,1934) shows that all three genes are expressed in many cell types in the hippocampus. The authors argue that the Cre lines (Supplementary Figure 1) show evidence that Akt2 is not expressed, however these blots are overexposed (and not quantified) and the lack of reduction in Cre lines will have limited sensitivity (I doubt if it would detect a 20% reduction). Thus, I would suggest the authors caution their statements "that Akt2 is not expressed in hippocampal neurons" (and cite these other sources of expression data) and that "Akt2 is specifically expressed in astrocytes". The authors could describe them as enriched in the particular cells.

3) Characterization of Akt inhibitors. Here the authors incubate brain slices with two different inhibitors and measure "the phosphorylation of Akt". Surely, it’s important to measure the phosphorylation of each individual Akt isoform? This could be achieved by immunoprecipitation of each form followed by phosphoAKT blotting. Have these inhibitors been tested on Akt1, 2 and 3?

4) The authors say that AZD inhibitor increases pAKT S473, however the blot in Figure 3 is not convincing (the 30 and 100micM bands look the same as vehicle).

5) The dissociation between the dose-dependent effects of MK and AZD on pAKTS473 and pGSK3bS9 is confusing. Why do both drugs inhibit pGSK3bS9 when MK blocks pAKTS473 and AZD increases pAKTS473? The authors refer to a paper (Okuzumi, et al., 2009) that describes an interesting phenomenon of "inhibitor hijacking" that may be relevant. The original paper does not describe this effect on the different Akt proteins and the experiments suggested in point 3 would also address this.

6) They decide to use the inhibitors at 30micM. This is a very high dose, especially when the MK drug strongly inhibits pAKTS473 at 10micM. ATP-competitive inhibitors are notorious for the poor specificity between families of kinases and it is typical to test the potential for non-specific inhibition of kinases. Using similar phospho-blots as in Figure 3, the authors could confirm that the doses they use are not interfering with other kinases that are known to interfere with LTP.

7) Figure 4 L-LTP inhibition. Figure 4 show the effects of the two inhibitors and from about 50' the drug-treated slices show a decremental LTP. In Supplementary Figure 2, the effect of the inhibitors on the baseline is only recorded for 30'. An important control that is missing is to record baseline for the same duration as the L-LTP experiments (200+ minutes). In the absence of this it is not possible to claim that "activity of AKT is required to sustain long-lasting LTP".

8) L-LTP experiment are technically very demanding and it is very easy to obtain decremental L-LTP simply as a result of poor health of the slice. To control for this, it is standard practice to perform 2-pathway experiments.

9) Since AKT inhibitors reduced L-LTP and pGSK3bS9, then it might be expected that Akt1-knockouts would also show reduced pGSK3bS9. But in Figure 6, apparently, this is not the case. Another dissociation is the reduction in pGSK3bS9 seen in Akt3 KO mice, which show normal L-LTP. This is confusing and important (the doses of the drugs were chosen on the basis that they inhibit pGSK3bS9).

10) Figure 5. Are statistics for slices or mice?

11) Figure 7, mGluR-LTD, would also benefit from 2-pathway experiments (they also exceed the 30 min baseline in the control Supplementary Figure 2.

12) The discrepancy between the results of LY294002 on mGluR-LTD is concerning. The original paper (Hou and Klann, 2004) showed they could reproduce the result with wortmannin. They also used an inactive structural analogue (LY294002), which had no effect. Given the controversial nature of this result, the authors should try these two other controls.

13) Figure 8, double mutant mice. This is a very nice line of investigation, especially since there is overlap of expression in Akt1 and Akt3. I am very surprised the authors have not performed the L-LTP experiment (and protein synthesis, and phosoph blots) to see if the partial effect on L-LTP in Akt1 kos is not further reduced in the double mutants.

Reviewer #3:

The is a solid paper with important new findings on Akt-isoform specific localization, signaling, and function in the hippocampus. The authors have effectively combined knockouts and conditional gene deletion strategies with pharmacology. The results are clear and the story is compelling and novel. Akt1, but not Akt3, supports activity-induced protein synthesis in L-LTP. Akt1 works in concert with Akt3 to inhibit mGluR-LTD, which is a completely unanticipated but well-documented finding. The article is well-written and the controversial findings, implications, and future directions are all discussed. I have only minor comments (below).

1) From the staining shown in Figure 1, it looks like Akt1 and Akt3 are expressed in the neuropil (synaptic/dendritic). The enzymes are not exclusively somatic. The authors should comment on this.

The synaptic/dendritic localization of Akt1 and Akt3 is an extremely important issue in view of the role of dendritic protein synthesis in synaptic maintenance and plasticity. PI3K/Akt signaling to mTOR has been shown to increase dendritic translation in CA1 by many labs. The authors mention the need for studies on localization but should spell out the need for work on Akt isoform signaling in local protein synthesis.

2) Figure 1. There appears to be a gap in Akt3 staining in the CA1 region. Is this representative or an artefact? This segment does not seem to correspond to CA2, but there is a lot of interest in CA2 recently. The authors could comment on the distribution of Akt in CA2.

3) The electrophysiology is from CA1. The authors should state in subsection “Akt1 deletion results in impaired protein synthesis after L-LTP induction”and the first paragraph of the Discussion section (and elsewhere, if applicable) that the findings on L-LTP/mGluR-LTD are from CA1.

4) Akt1 and Akt3 are shown to have different roles in protein synthesis, and there appears to be a differential distribution of the enzymes in CA1, CA3, and DG. It is notable in this context that rapamcyin blocks L-LTP and CA1 but does not affect L-LTP in the DG (Panja et al., 2009, 2014). The regional differences in L-LTP and translation might be related to differential activation of Akt isoforms.

With regard to the 2 pathway experiments raised by Reviewer #2, we feel that this could be addressable if the AKT inhibitors do not cause baseline synaptic transmission to run down over the course of the 1-2 hour duration of the L-LTP experiments. Currently, the effect of the inhibitors is only during a 30 min application.

---

## [Author Response]

Reviewer #1:1) I commend the authors for doing a thorough job characterizing the AKT inhibitors in the slice preparation in Figure 3. However, there are some confusing results that may require more explanation. The MK inhibitor at 10um completely blocks P-AKT, but has no effect on its substrate GSK3B, whereas the AZD drug enhanced P-AKT, but inhibits GSK3B phosphorylation. Is AKT autophosphorylated? Or are the distinct effects of the inhibitors due to feedback onto PDK1? It would be good to see the same data in the AKT1 and AKT3 KO mice to see if similar or different effects are observed.

Originally, we were also surprised by the AZD result but then found that several studies have investigated this phenomenon. This inhibitor-induced “paradoxical” AKT hyperphosphorylation is not due to enhancement of upstream signals to compensate for the AKT signal loss but rather is related to the occupation and binding of this type of inhibitor at the ATP-pocket of AKT. Therefore, the hyperphosphorylation of AKT following AZD is not due to a feedback loop but due to ‘intrinsic’ mechanisms. Previously, it was shown that binding of the ATP-competitive inhibitor results in translocation of AKT to the membrane (Okuzumi et al., 2009). Here at the membrane it seems that AKT is either more susceptible to phosphorylation (Okuzumi et al., 2009) or more resistant to dephosphorylation (Chan et al., 2011), but either way AKT activity is inhibited. We have now explained this in more detail in the paper. Nevertheless, to make sure there was no feedback loop in brain slices, we tested phosphorylation of PDK1 and found no effect of AZD or MK (Figure 3—figure supplement 1). The inhibitory effects of AZD and MK on AKT activity have been tested and verified previously, showing differences in potency, consistent with our dose-dependent GSK3β results (MK2206: AKT_1/2_/3 with IC50 8/12/65 nM^1^; AZD5363: AKT_1/2_/3 with IC50 3/8/8 nM^2^. We also tested the effect of both inhibitors on another AKT substrate, TSC2 and found that this substrate was inhibited as well. Finally, we tested the effect of the inhibitors on each isoform and found that MK2206 dephosphorylates all isoforms and AZD hyperphosphorylates all isoforms. Based on these results, we believe it is unlikely that the drugs will perform differently in the context of an Akt mutant. Additionally, because we did not report any results combining pharmacology with Akt deficient mice, we believe testing the AKT inhibitors in Akt mutants will not provide any significant new information relative to the data already presented.

2) In Figure 4, a pharmacological and genetic approach is used to study the effect of AKT on L-LTP. The Akt1 isoform is claimed to be the one involved in the maintenance of L-LTP and the target of the pharmacological blockers. Is somehow surprising that in AKT1 KO, the L-LTP seems to be only partially inhibited, whereas both of the pharmacological agents completely block this form of synaptic plasticity. Is the effect of the blockers specific? What is the effect of these inhibitors on the LTP induced in the AKT1 knockout mice.

We agree that both inhibitors block L-LTP stronger compared to Akt1 KO alone. This could suggest that another isoform has an effect as well, most likely AKT3. The absence of an L-LTP impairment with Akt3 KO suggests that AKT1 can compensate for the loss of AKT3 or that AKT1 is the predominant kinase involved in this form of synaptic plasticity. Indeed, our data support the idea that AKT1, but not AKT3, is specifically involved in protein synthesis induced by high frequency stimulation (Figure 5). To more thoroughly explore this question, the cA1FA3K double mutants would be the best model to answer this question, and we have planned to examine L-LTP in this model for future studies. However, we believe this is beyond the scope of the current study because it is primarily focused on the role of single isoforms in synaptic plasticity. Another detail, as may be noticed from Figure 8, is that the double mutants are rather difficult to generate (1 in 8 mice, or 1 mouse per 2 litters, will be a male double mutant); therefore, we chose instead to use the mice generated in our revision time frame for slices to answer point 5 made by this reviewer.

3) In Figure 5, the authors show impaired protein synthesis in AKT1 KO after L-LTP induction. Are there any differences in protein synthesis rate under basal condition in each of the KO mice? That is not reported and would help to interpret the results.

We agree this would be very interesting. In the current paper, we controlled for basal protein synthesis differences between animals by performing the puromycin-labeling experiments within each animal, meaning that slices obtained from a singleanimal were either not stimulated or stimulated using HFS. The stimulated slices were then compared to the unstimulated slices for each animal to examine protein synthesis rates. This was repeated for each genotype separately. This approach is necessary because basal labeling rates can vary between different animals. Several factors can account for this, such as recent metabolism stimulation (i.e. feeding), sleep, and other factors that are difficult to control for across individual mice. Therefore, even though this would be a great question to answer, other strategies might be better used to address it. In future studies, we plan to employ a luciferase translation reporter to measure synthesis rates both in culture and in vivo.

4) In Figure 5, the effect of HFS on S6 phosphorylation in AKT1 and AKT3 ko is reported. The phosphorylation of S6 after HFS does not look different in the wild type slice, perhaps because they are saturated. What seems to be different is only the total amount of S6 that is reduced in the WT after HFS. It would be nice to see a loading control, actin or tubulin?, for these western. Also in this case, knowing the effect of AKT1 and AKT3 absence on basal level of S6 phosphorylation would help to interpret the results.

At the reviewer’s suggestion, we have included loading controls (GAPDH) for these westerns. We agree with the reviewer that the enhanced pS6 in WT samples after HFS seems largely due to reduced total S6 levels. We observed a similar phenomenon in an earlier study (3). So, while we agree that a simpler result would have been increased pS6 with no change in total S6 levels, it appears that consistent with previous results pS6 increases are driven by changes in total S6 protein. A previous study has also shown that basal pS6 levels are reduced in *Akt3* KO mice but not affected in *Akt1* KO mice (4). We observed the same results but because this was known and we were interested in how the *Akt* mutants responded to HFS, we had chosen to present our data to focus on their protein synthesis response post-HFS for clarity considerations. However, to address this point, we have now added discussion about pS6 levels basally.

5) The most surprising, and therefore interesting, result in the paper, in my opinion, is Figure 7, where the authors show enhanced mGlu-LTD using the inhibitors of Akt, and this result is confirmed with PI3K inhibitors and in AKT1/AKT3 CA1 dKO. The authors discuss a role for AKT as a brake on mGlu-LTD and protein synthesis machinery induction. However, they present no data to support this conclusion. Because these authors used SunSET to examine effects of AKT deletion on HFS-induced protein synthesis, this should also be done with mGluR-LTD. This result may suggest that AKT plays distinct roles in protein synthesis regulation depending on the stimulus (LTP or LTD), which in my opinion is very novel and impactful. Also, examining the effects of the inhibitors and AKT deletion on DHPG-induced signaling to mTORC1 and other pathways would provide more mechanistic insight into how AKT does this. However, this could be beyond the scope of this present manuscript.We agree with the reviewer that this is of great interest and therefore attempted to address this point with new data in Figure 8. Intriguingly, these results show that TSC2 signaling, upstream of mTORC1, is not altered whereas ERK signaling is enhanced in the double KO mice 60 min after DHPG treatment, leading to increased p-S6 signaling as well. This suggests that protein synthesis is indeed enhanced in cA1FA3K mice and that AKT1 and AKT3 together normally act as a brake on mGluR-LTD and the underlying protein synthesis. Owing to material (slice availability) and time constraints, we did not perform SuNSET analyses with the double mutant cA1F/A3K mice. The SuNSET protocol in our hands requires nearly all the slices generated from a single mouse to provide sufficient materials for control, stimulated and unlabeled experimental samples with technical replicates.Reviewer #2:1) The introduction to the Akt family is confusing and non-standard with respect to genes, gene families and proteins. The authors refer to "three highly related isoforms". There are many splice isoforms described for each of three genes (see Ensemble and UniProt). Nowhere to they explain that they are referring to three genes, and each one expresses multiple splice isoforms. A figure with relevant schematic sequence alignments of the proteins and explanation of the isoforms and paralogs would be useful, showing where the antibodies bind and where the location of domains that bind to the drugs. This would enable the reader to understand the limitations of the specificity of the reagents.

We apologize that the background on the AKT family was confusing. The manuscript now explains in more detail that we are examining three AKT protein isoforms and each isoform is generated from a different gene (highlighted). While indeed Ensembl describes multiple splice isoforms for each of the three AKT genes, only the mRNA transcripts are different, and all transcripts of a gene result in the same protein. For example, human AKT1 has six mRNA splice isoforms (208, 202, 214, 201, 203, 211) which differ in their 5’ or 3’ untranslated region (UTR) but all result in the same protein. Although these UTR differences may be important for mRNA translation or localization, the protein is the same. Ensembl describes four other AKT1 splice variants (207, 213, 215, 209), which are shorter peptides, but these isoforms are “derived from an Ensembl automatic analysis pipeline and should be considered as preliminary data” (Uniprot). In addition, these isoforms have no CCDS code and therefore may not even exist. The CCDS project is a collaborative effort to identify a core set of protein coding regions that are consistently annotated and of high quality, so isoforms lacking such a code are not considered high-quality and may not be biologically relevant. In mice, the Akt1 gene has one isoform that is confirmed (201), while the other shorter peptides (205, 208, 206) are “derived from an Ensembl automatic analysis pipeline and should be considered as preliminary data” (Uniprot). The Akt2 gene has four splice isoforms (205, 204, 201, 214) that differ in their 5’ or 3’ UTR but result in the same protein. The other Akt2 splice isoforms are derived from an Ensembl automatic analysis pipeline but not verified. Only the Akt3 gene has a small difference in the last exon, where the Serine 472 phospho-site is found for the main isoforms 202 and 203, while the other isoform 201 has a different C-terminal. However, the 201 isoform has no CCDS code, suggesting it may not be a high-quality transcript. As with Akt1 and Akt2, all other Akt3 isoforms are predicted but not verified. Considering this information, we do not think the manuscript will benefit from describing mRNA splice variants because those verified for an Akt gene lead to the same protein and the focus of our study is the function of the three AKT protein isoforms in synaptic plasticity, not where the mRNA is localized or to what degree it is translated.

We had shared the same concern as the reviewer about the specificity of our reagents, which is why we used more than one AKT inhibitor in addition to genetic approaches. We also included staining in knockout samples to confirm specificity of our AKT antibodies. Although the exact epitope of these antibodies was confidential, we could retrieve information about the central peptide used for each antibody. To the best of our knowledge, these epitopes would be present in all Akt splice variants. Therefore, isoforms of each Akt gene (should they exist) would be recognized by our molecular tools. Our western results reveal single bands for both phospho- and total AKT proteins, consistent with the notion that each Akt gene generates one protein isoform. However, to address the reviewer’s concern, we have added epitope information to the Materials and methods section.

2) The differential expression of the three genes shown using immunolabelling in Figure 1 appears to describe differential expression, with Akt2 showing most differences to Akt1 and 3. An inspection of expression data in the Allen References Atlas shows Akt1,2 and 3 are expressed in all principal neurons, but Ak2 is at lower levels than either Akt1 and 3. Inspection of single-cell transcriptome data (Zeisel et al.,1934) shows that all three genes are expressed in many cell types in the hippocampus. The authors argue that the Cre lines (Supplementary Figure 1) show evidence that Akt2 is not expressed, however these blots are overexposed (and not quantified) and the lack of reduction in Cre lines will have limited sensitivity (I doubt if it would detect a 20% reduction). Thus, I would suggest the authors caution their statements "that Akt2 is not expressed in hippocampal neurons" (and cite these other sources of expression data) and that "Akt2 is specifically expressed in astrocytes". The authors could describe them as enriched in the particular cells.

We appreciate the reviewer’s caution, but we have tested many different AKT2 antibodies, of which most were not specific (showing staining in the Akt2 KO). Once we found an antibody that was specific, we were also initially surprised that the immunostaining for AKT2 showed high expression in astrocytes only. That is why we proceeded with experiments to target AKT2 selectively using Cre-mediated removal. We were also aware that the Allen Brain Atlas showed Akt2 transcripts in most cells. However, this is mRNA expression data, and mRNA that are produced do not necessarily get translated into protein. Ironically, the Zeisel et al. study cited by the reviewer reports the presence of *only* Akt2 in astrocytes and not the other isoforms, which is in fact consistent with our data (See Zeisel et al., Supplementary table S1).

Although our western blot data were collected in the linear range of detection, we have replaced the AKT2 images with lower exposures and quantified the signals to address the reviewer’s concern. In addition, we added new AKT2 removal data using Nestin-driven Cre. Nestin is expressed early during embryogenesis in neural progenitor cells that become astrocytes and neurons. We found that Nestin-Cre mediated complete abolishment of AKT2 expression in the hippocampus (Figure 2—figure supplement 1), whereas neuron-specific CamkIIa-Cre or NSE-Cre did not appear to remove any AKT2 protein, supporting our idea that AKT2 is expressed specifically in astrocytes. However, because we cannot rule out trace amounts of AKT2 in neurons, we have softened our conclusion to say, “our data suggests that AKT2 is not expressed in hippocampal neurons but is mainly expressed in astrocytes” and have included discussion of other AKT expression data.

3) Characterization of Akt inhibitors. Here the authors incubate brain slices with two different inhibitors and measure "the phosphorylation of Akt". Surely, it’s important to measure the phosphorylation of each individual Akt isoform? This could be achieved by immunoprecipitation of each form followed by phosphoAKT blotting. Have these inhibitors been tested on Akt1, 2 and 3?

These inhibitors have been validated and studied previously and their pharmacological effects on each isoform have been tested (MK 2206: Akt_1/2_/3 with IC50 8/12/65 nM (1); AZD5363 Akt_1/2_/3 with IC50 3/8/8 nM (2). Our own independent data in brain slices to determine what dose to use verified the effects of these AKT inhibitors (Figure 3 and Figure 3—figure supplement 1). However, as the reviewer suggested, we have now added data showing the effect of both inhibitors on the phosphorylation status of each isoform (Figure 3—figure supplement 1). In agreement with our pan-AKT results, AZD5363 incubation hyperphosphorylates all AKT isoforms while MK2206 results in reduced phosphorylation of all AKT isoforms.

4) The authors say that AZD inhibitor increases pAKT S473, however the blot in Figure 3 is not convincing (the 30 and 100micM bands look the same as vehicle).

We apologize for the selected image and in response to this critique, we replaced the blot with a more representative image.

5) The dissociation between the dose-dependent effects of MK and AZD on pAKTS473 and pGSK3bS9 is confusing. Why do both drugs inhibit pGSK3bS9 when MK blocks pAKTS473 and AZD increases pAKTS473? The authors refer to a paper (Okuzumiet al., 2009) that describes an interesting phenomenon of "inhibitor hijacking" that may be relevant. The original paper does not describe this effect on the different Akt proteins and the experiments suggested in point 3 would also address this.

We apologize that this was not presented clearly in the original manuscript. Please see our response to point 3 as well as our response to point 1 from the first reviewer, who was similarly confused by the paradoxical effect of the two inhibitors on phosphorylation of AKT.

6) They decide to use the inhibitors at 30micM. This is a very high dose, especially when the MK drug strongly inhibits pAKTS473 at 10micM. ATP-competitive inhibitors are notorious for the poor specificity between families of kinases and it is typical to test the potential for non-specific inhibition of kinases. Using similar phospho-blots as in Figure 3, the authors could confirm that the doses they use are not interfering with other kinases that are known to interfere with LTP.

We understand the concern, which is noteworthy for all pharmacological studies. For this reason, our study also included genetic removal of AKT activity, although this approach carries its own caveats. Together however, the two approaches allow us to generate converging evidence in support of our conclusions. The congruent findings from our pharmacological and genetic studies strongly suggest that we were not observing non-specific effects of either approach. In addition, previous studies have used the AKT inhibitors at higher concentrations and longer durations than our study. For example, studies have reported inhibition of AKT by ~1-3 µM AZD applied up to 36 h on cells grown in a monolayer (5, 6) but in vivo application required ~69 mM for an adult mouse (200 mg/kg) (6). For MK2206, in vitro doses that have been used range between 1-20 µM for 2-72 h for cell monolayers (7-9) while in vivo application required ~75 mM for an adult mouse (240 mg/kg) (1). The brain slices that are used in our study are 400 µm thick, consisting of multiple cell layers, and treated for a short amount of time (30 min). Therefore, we disagree that 30 µM is a high dose, especially as it resulted in similar effects as genetic AKT removal. Furthermore, this was the *minimum* dose required to block substrate activation. Using less would have left incomplete AKT blockade, confounding interpretation of our results. Finally, we have tested phosphorylation of PDK in response to reviewer 1, but the results also apply here, showing that the inhibitor doses are not interfering with at least one other kinase. To more extensively address the question of potential off-target effects, an unbiased kinome screen using a mass spec approach would be required, which is well beyond the scope of this study.

7) Figure 4. L-LTP inhibition. Figure 4 show the effects of the two inhibitors and from about 50' the drug-treated slices show a decremental LTP. In Supplementary Figure 2, the effect of the inhibitors on the baseline is only recorded for 30'. An important control that is missing is to record baseline for the same duration as the L-LTP experiments (200+ minutes). In the absence of this it is not possible to claim that "activity of AKT is required to sustain long-lasting LTP".

Although the L-LTP impairment in *Akt1* KO slices supports our inhibitor results, we performed longer baseline recordings following inhibitor treatment at the reviewer’s request. We found no effect of AZD or MK on baseline recordings for as long as 90 min post-AKT inhibitor incubation (Figure 4—figure supplement 1).

8) L-LTP experiment are technically very demanding and it is very easy to obtain decremental L-LTP simply as a result of poor health of the slice. To control for this, it is standard practice to perform 2-pathway experiments.

We respectfully disagree that 2-pathway experiments are standard practice. L-LTP induced by high frequency stimulation is a well-established paradigm, for which 2-pathway experiments in the early days have confirmed that changes in synaptic plasticity are a phenomenon that occurs only after stimulation of a specific pathway. Most studies in mouse hippocampal slices have since not included 2-pathway experiments. Additionally, the reviewer seems to suggest that our results are due to poor health of Akt1 KO slices and AZD- and MK-treated slices only in the context of L-LTP and mGluR-LTD experiments; we do not believe this is a fair assessment. However, to address this critique, we have performed the baseline stability experiments in which slices were incubated with the inhibitors for extended periods of time (see point 7). These results show that (1) we can produce slices healthy enough to maintain stable baselines and (2) the inhibitors themselves do not reduce baseline fEPSP responses.

9) Since AKT inhibitors reduced L-LTP and pGSK3bS9, then it might be expected that Akt1-knockouts would also show reduced pGSK3bS9. But in Figure 6, apparently, this is not the case. Another dissociation is the reduction in pGSK3bS9 seen in Akt3 KO mice, which show normal L-LTP. This is confusing and important (the doses of the drugs were chosen on the basis that they inhibit pGSK3bS9).

We agree that the GSK3β results and their relationship to L-LTP are complicated, but these dissociations are the essence of our study. We had hoped to reveal signaling mechanisms linked specifically to different AKT isoforms and it appears GSK3β is one of them. GSK3β is a well-known and validated AKT target whose activity is blocked when it is phosphorylated by AKT (unphosphorylated GSK3β is the active form) (10) and was therefore chosen to validate our drug concentration. While it would have been more convenient for our study if Akt1 KO also led to reduced pGSKβ levels as with the AKT inhibitors, the data are reproducibly what we have presented. In Akt1 KO hippocampal tissue, pGSK3β levels are normal whereas in Akt3 KO tissue, pGSK3β is reduced (GSK3β is more active). We think the simplest explanation for these results is that specific AKT isoforms have preferences for certain targets, but other isoforms may be able to compensate. Thus, our results may suggest that the AKT3 isoform is specifically important for regulating pGSK3β, but reduced pGSK3β does not necessarily result in impaired L-LTP. Previously, it was found that L-LTP results in increased pGSK3β (less active GSK3β) (11). Therefore, it is possible that the presence of AKT inhibitors prevented GSK3β from being modified, resulting in impaired L-LTP. On the other hand, that GSK3β is more active in Akt3 KO yet L-LTP is not affected suggests either that AKT1 could compensate for AKT3 loss or that pGSK3β is not the mechanism mediating impaired L-LTP (Figure 5). Indeed, the revised manuscript now includes data on the phosphorylation of TSC2 (Figure 8) to show that the inhibitors affect other downstream AKT targets as well, which justifies our drug concentration but also suggests that several other AKT mechanisms besides GSK3β regulation could underlie L-LTP. We agree with the reviewer that these are important points, so we have expanded discussion in the manuscript to help with clarity and interpretation of these results.

10) Figure 5. Are statistics for slices or mice?

We apologize for the oversight and have clarified this section.

11) Figure 7, mGluR-LTD, would also benefit from 2-pathway experiments (they also exceed the 30 min baseline in the control Supplementary Figure 2.

Please see point 7. As discussed also for point 8, we respectfully disagree that 2-pathway experiments should be performed, especially in the case of mGluR-LTD induced by DHPG, which is applied to the bath.

12) The discrepancy between the results of LY294002 on mGluR-LTD is concerning. The original paper (Hou and Klann, 2004) showed they could reproduce the result with wortmannin. They also used an inactive structural analogue (LY294002), which had no effect. Given the controversial nature of this result, the authors should try these two other controls.

Although the 2004 paper indeed aimed to address the role of AKT in mGluR-LTD, it did not directly block AKT activity or use genetic animal models to confirm the results. Instead, the role of AKT was *inferred* from upstream inactivation of PI3K, whereas our study *directly* targets AKT using pharmacological and genetic approaches. Our results were unexpected based on the 2004 paper but not inconceivable because upstream manipulations can alter other pathways parallel to AKT signaling that lead to mGluR-LTD, which our study aimed to dissect by directly assessing AKT activity. Using three different compounds to inhibit AKT activity and a neuron-specific AKT1/3 double mutant mouse, we consistently found enhanced DHPG-induced mGluR-LTD. We did not perform mGluR-LTD experiments with wortmannin or the inactive analogue of LY294002 because the focus of our study was downstream and not upstream of AKT. We agree with the reviewer that the conflicting LY294002 results from the 2004 study and ours is concerning, but the preponderance of data suggest AKT activity is a brake on mGluR-LTD. On balance, we think the strength of our tools (multiple AKT inhibitors and genetic AKT removal) provides strong support for our conclusions, despite potential controversies generated here. We hope in the future that another group can independently confirm our results so that the role of AKT in this form of synaptic plasticity can be better clarified.

13) Figure 8, double mutant mice. This is a very nice line of investigation, especially since there is overlap of expression in Akt1 and Akt3. I am very surprised the authors have not performed the L-LTP experiment (and protein synthesis, and phosoph blots) to see if the partial effect on L-LTP in Akt1 kos is not further reduced in the double mutants.

We enthusiastically agree with the reviewer that this would be a very exciting line of investigation, and we are planning to do these experiments in a follow-up paper. However, as we explained for reviewer 1, since these mutant mice are difficult to generate, we chose to use our available double mutant mice to better understand the signaling mechanisms behind enhanced mGluR-LTD after AKT blockade for the current study. In the future, we will investigate these mice in greater detail.

Reviewer #3:The is a solid paper with important new findings on Akt-isoform specific localization, signaling, and function in the hippocampus. The authors have effectively combined knockouts and conditional gene deletion strategies with pharmacology. The results are clear and the story is compelling and novel. Akt1, but not Akt3, supports activity-induced protein synthesis in L-LTP. Akt1 works in concert with Akt3 to inhibit mGluR-LTD, which is a completely unanticipated but well-documented finding. The article is well-written and the controversial findings, implications, and future directions are all discussed. I have only minor comments (below).1) From the staining shown in Figure 1., it looks like Akt1 and Akt3 are expressed in the neuropil (synaptic/dendritic). The enzymes are not exclusively somatic. The authors should comment on this.The synaptic/dendritic localization of Akt1 and Akt3 is an extremely important issue in view of the role of dendritic protein synthesis in synaptic maintenance and plasticity. PI3K/Akt signaling to mTOR has been shown to increase dendritic translation in CA1 by many labs. The authors mention the need for studies on localization but should spell out the need for work on Akt isoform signaling in local protein synthesis.

At the reviewer’s suggestion, we have addressed these points in the Discussion section.

2) Figure 1. There appears to be a gap in Akt3 staining in the CA1 region. Is this representative or an artefact? This segment does not seem to correspond to CA2, but there is a lot of interest in CA2 recently. The authors could comment on the distribution of Akt in CA2.

It may seem like an artefact but is representative of most of our AKT3 staining images in the WT hippocampus. We have included a different AKT3 staining image here as another example for the reviewer. We have also commented on AKT distribution in CA2 in the revised manuscript, as the reviewer suggested.

3) The electrophysiology is from CA1. The authors should state in subsection “Akt1 deletion results in impaired protein synthesis after L-LTP induction”and the first paragraph of the Discussion section (and elsewhere, if applicable) that the findings on L-LTP/mGluR-LTD are from CA1.

We have added this detail to the first paragraph of the Discussion section and in other places of the text.

4) Akt1 and Akt3 are shown to have different roles in protein synthesis, and there appears to be a differential distribution of the enzymes in CA1, CA3, and DG. It is notable in this context that rapamcyin blocks L-LTP and CA1 but does not affect L-LTP in the DG (Panja et al., 2009, 2014). The regional differences in L-LTP and translation might be related to differential activation of Akt isoforms.

This is an excellent point and we have included it in our Discussion section.

References:

1. Yan, L. MK-2206: A potent oral allosteric AKT inhibitor. Cancer Research 69 (2009).

2. Addie, M., et al. Discovery of 4-amino-N-[(1S)-1-(4-chlorophenyl)-3-hydroxypropyl]-1-(7H-pyrrolo[2,3-d]pyrimidin -4-yl)piperidine-4-carboxamide (AZD5363), an orally bioavailable, potent inhibitor of Akt kinases. J Med Chem 56, 2059-2073 (2013).

3. Antion, M.D., et al. Removal of S6K1 and S6K2 leads to divergent alterations in learning, memory, and synaptic plasticity. Learn Mem 15, 29-38 (2008).

4. Easton, R.M., et al. Role for Akt3/protein kinase Bgamma in attainment of normal brain size. Mol Cell Biol 25, 1869-1878 (2005).

5. Jang, K.J., et al. Mitochondrial function provides instructive signals for activation-induced B-cell fates. Nat Commun 6, 6750 (2015).

6. Davies, B.R., et al. Preclinical pharmacology of AZD5363, an inhibitor of AKT:

pharmacodynamics, antitumor activity, and correlation of monotherapy activity with genetic background. Mol Cancer Ther 11, 873-887 (2012).

7. Sefton, E.C., et al. MK-2206, an AKT inhibitor, promotes caspase-independent cell death and inhibits leiomyoma growth. Endocrinology 154, 4046-4057 (2013).

8. Zhang, L., et al. Microenvironment-induced PTEN loss by exosomal microRNA primes brain metastasis outgrowth. Nature 527, 100-104 (2015).

9. Vivanco, I., et al. A kinase-independent function of AKT promotes cancer cell survival. eLife (2014).

10. Cross, D.A., Alessi, D.R., Cohen, P., Andjelkovich, M. & Hemmings, B.A. Inhibition of glycogen synthase kinase-3 by insulin mediated by protein kinase B. Nature 378, 785-789 (1995).

11. Peineau, S., et al. LTP inhibits LTD in the hippocampus via regulation of GSK3beta. Neuron 53, 703-717 (2007).